# Radiocarbon and genomic evidence for the survival of *Equus Sussemionus* until the late Holocene

**Dawei Cai[1]\*[†], Siqi Zhu[1][†], Mian Gong[2][†], Naifan Zhang[1], Jia Wen[2], Qiyao Liang[1], Weilu Sun[1], Xinyue Shao[1], Yaqi Guo[1], Yudong Cai[2], Zhuqing Zheng[2], Wei Zhang[3], Songmei Hu[4], Xiaoyang Wang[5], He Tian[3], Youqian Li[3], Wei Liu[3], Miaomiao Yang[4], Jian Yang[5], Duo Wu[6], Ludovic Orlando[7]\*, Yu Jiang[2]\***

[1]Bioarchaeology Laboratory, Jilin University, Changchun, China; [2]Key Laboratory of Animal Genetics, Breeding and Reproduction of Shaanxi Province, College of Animal Science and Technology, Northwest A&F University, Yangling, China; [3]Heilongjiang Provincial Institute of Cultural Relics and Archaeology, Harbin, China; [4]Shaanxi Provincial Institute of Archaeology, Xi'an, China; [5]Ningxia Institute of Cultural Relics and Archaeology, Yinchuan, China; [6]College of Earth and Environmental Sciences; MOE Key Laboratory of Western China's Environmental Systems, Lanzhou University, Lanzhou, China; [7]Centre d'Anthropobiologie et de Génomique de Toulouse (CAGT), CNRS UMR 5288, Université de Toulouse, Université Paul Sabatier, Toulouse, France, Toulouse, France

**\*For correspondence:**
caidw@jlu.edu.cn (DC);
ludovic.orlando@univ-tlse3.fr (LO);
yu.jiang@nwafu.edu.cn (YJ)

[†]These authors contributed equally to this work

**Competing interest:** The authors declare that no competing interests exist.

**Abstract** The exceptionally rich fossil record available for the equid family has provided textbook examples of macroevolutionary changes. Horses, asses, and zebras represent three extant subgenera of *Equus* lineage, while the *Sussemionus* subgenus is another remarkable *Equus* lineage ranging from North America to Ethiopia in the Pleistocene. We sequenced 26 archaeological specimens from Northern China in the Holocene that could be assigned morphologically and genetically to *Equus ovodovi*, a species representative of *Sussemionus*. We present the first high-quality complete genome of the *Sussemionus* lineage, which was sequenced to 13.4× depth of coverage. Radiocarbon dating demonstrates that this lineage survived until ~3500 years ago, despite continued demographic collapse during the Last Glacial Maximum and the great human expansion in East Asia. We also confirmed the *Equus* phylogenetic tree and found that *Sussemionus* diverged from the ancestor of non-caballine equids ~2.3–2.7 million years ago and possibly remained affected by secondary gene flow post-divergence. We found that the small genetic diversity, rather than enhanced inbreeding, limited the species' chances of survival. Our work adds to the growing literature illustrating how ancient DNA can inform on extinction dynamics and the long-term resilience of species surviving in cryptic population pockets.

## Editor's evaluation

This article represents multiple milestones in our understanding of the evolution and extinction of Pleistocene equids, including revising the timing of extinction and clarifying the evolutionary history of *Equus* (*Sussemionus*) *ovodovi*. The discovery of the late persistence of non-caballine equid taxa in Northern China until deep into the late Holocene is particularly important. This finding will be of broad interest to the paleontology, paleoecology, archaeology, and paleogenomic communities and should stimulate important future research into equid extinction processes.

## Introduction

Today, all of the seven extant species forming the horse family belong to one single genus, *Equus*. It emerged in North America some 4.0–4.5 million years ago (*Orlando et al., 2013*), and first spread into Eurasia ~2.6 million years ago (Mya), via the Beringia land bridge (*Lindsay et al., 1980*). This first vicariance and expansion out of America led to the emergence of the ancestors of zebras, hemiones, and donkeys, a group collectively known as non-caballine (or stenonine) equids. Another expansion through Beringia occurred ~0.8–1.0 Mya (*Vershinina et al., 2021*), which allowed caballine equids (i.e., those most closely related to the horse) to enter into the Old World, where they persisted until the modern era and were domesticated ~5500–4200 years ago (*Gaunitz et al., 2018*; *Librado et al., 2021*; *Outram et al., 2009*).

In the recent years, ancient DNA (aDNA) data have revealed that the genetic diversity of non-caballine *Equus* was considerably larger in the past than it is today (*Librado and Orlando, 2021*; *Orlando, 2020*). This was further confirmed as the first mitochondrial DNA (mtDNA) data of *Equus* (*Sussemionus*) were collected (hereafter referred to as Sussemiones) (*Eisenmann, 2010*). This lineage radiated across North America, Africa, and Siberia, and developed multiple adaptations to a whole range of arid and humid environments (*Eisenmann, 2010*). Sussemiones were first believed to have become extinct during the Middle Pleistocene as the last known specimen showing typical morpho-anatomical characters dated back to ~500 kya (thousand years ago) in southeastern Siberia, Russia (*Vasiliev, 2013*). However, DNA results obtained on multiple osseous remains within the radiocarbon range and showing morphological traits reminiscent of the Eurasian Sussemiones species indicated that the lineage in fact survived until the Late Pleistocene (*Druzhkova et al., 2017*; *Orlando et al., 2009*; *Vilstrup et al., 2013*; *Yuan et al., 2019*). Early publications indicated survival dates 40–50 kya in southeastern Siberia, Russia (Proskuryakova cave) (*Orlando et al., 2009*; *Vilstrup et al., 2013*), ~32 kya at the Denisova cave (*Druzhkova et al., 2017*), and ~12.6 kya at northeastern China (*Yuan et al., 2019*).

Despite an abundant fossil material, only a limited number of Sussemiones specimens have been investigated for ancient mitochondrial DNA (aDNA). These studies showed that Sussemiones formed a non-caballine lineage that may have diverged first from the lineage ancestral to zebras, hemiones, and donkeys. However, the exact placement of Sussemiones could not be fully resolved and has remained contentious (*Heintzman et al., 2017*; *Orlando et al., 2009*; *Vilstrup et al., 2013*). In this study, we have carried out archaeological excavations in three Holocene sites in China (Honghe, Heilongjiang Province; Muzhuzhuliang, Shaanxi Province; Shatangbeiyuan, Ningxia Province) (*Figure 1* and *Supplementary file 1a*) and uncovered equine samples showing morphological features that may be characteristic of Sussemiones.

Equine assemblages dating to prior to the late Shang dynasty (ca. 3300 years ago) have documented the presence of wild horses in Northern China during the Late Pleistocene (*Yuan and Flad, 2006*). The taxonomic status and/or stratigraphic placement of the rare material attributed to Neolithic and early Shang contexts remained, however, contentious, leaving the possibility that Sussemiones or other equid taxa (co-)existed in China at the time, especially at the sites investigated in this study. At the Honghe site (47.20°N, 123.62°E), excavation fieldwork of nearly 20,000 m$^2$ has uncovered a late Neolithic settlement site dated to ~3400–4400 years ago, which belonged to a unique, rich fishing and hunting culture characteristic of northeastern China (*Figure 1—figure supplement 1*). The scale of the moated settlement indicated that there was already social management and relatively high productivity and building technology (*Zhang et al., 2020*). The Muzhuzhuliang site (38.83°N, 110.50°E) belonged to the '*Longshan culture*.' It was dated to ~3800–4300 years ago and represents the most complete moated settlement hitherto excavated in the late Neolithic age of Northern China, showing a mixed subsistence economy involving agriculture, animal husbandry, and hunting (*Wang et al., 2015*). Finally, the Shatangbeiyuan site (35.63°N, 105.11°E) belonged to the early cultural relics of Neolithic '*Qijia culture*,' which was dated to ~3900–4200 years ago. While millet represented the main crop produced at that time, findings including stone and bone arrowheads have also supported the presence of hunting (*Fan et al., 2017*). No obvious signs of domestication, including paleopathologies related to horseback riding, bridling, or chariotry (*Bendrey, 2007*; *Taylor and Tuvshinjargal, 2018*), were found amongst the equine specimens investigated at the three sites. In contrast, slash marks could be identified on some of the bones (HH13H, HH26H, and MZ104H), together with indications of bone marrow extraction (*Figure 1—figure supplement 2*). These findings suggest that these specimens were hunted.

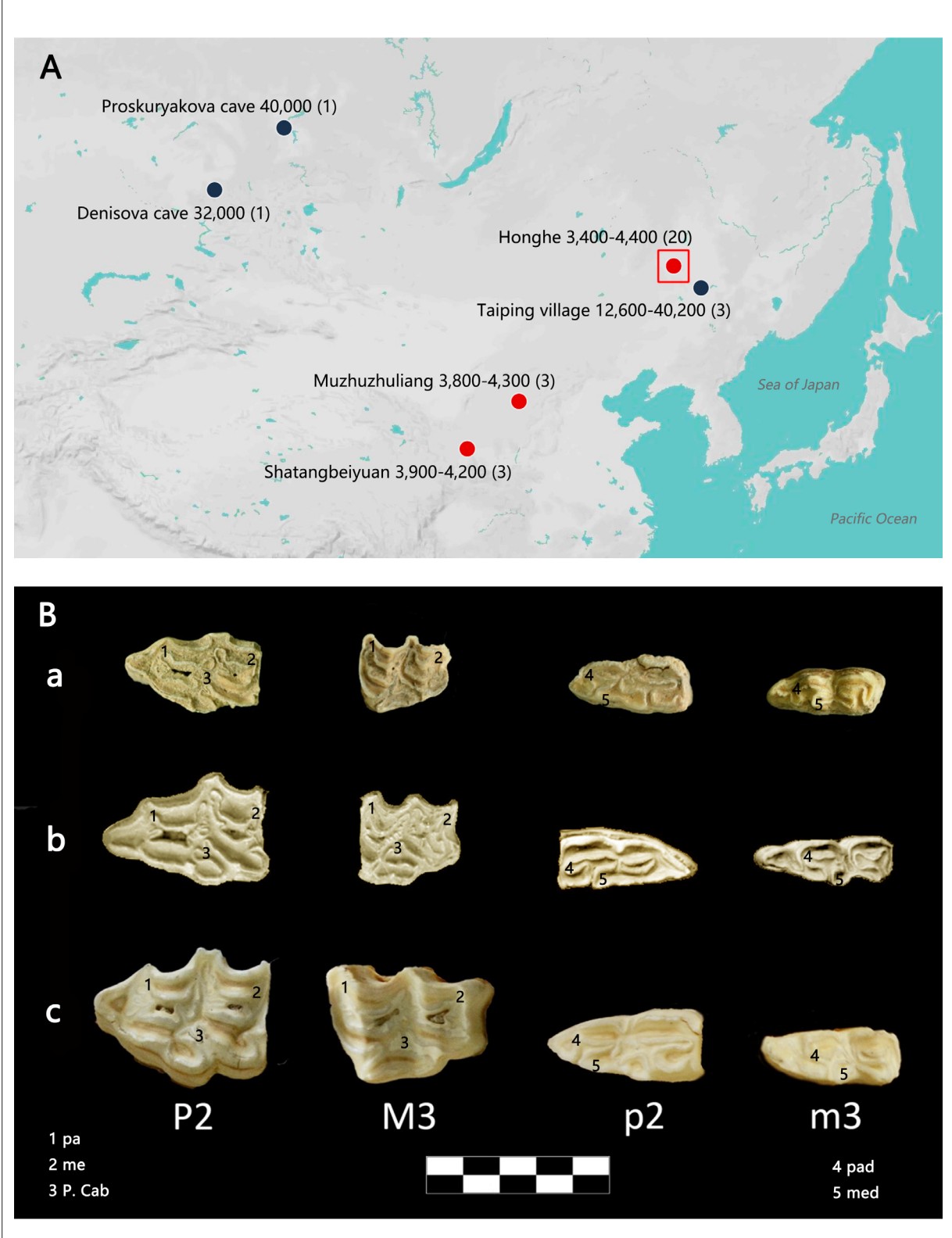

**Figure 1.** Geographic distribution of *E.* (*Sussemionus*) *ovodovi* specimens investigated at the DNA level and pre-molar and molar morphology. (**A**) *E.* (*Sussemionus*) *ovodovi* geographic range. The three red circles indicate the archaeological sites analyzed in this study. The site (Honghe) that delivered the complete genome sequence at 13.4-fold average depth of coverage (HH06D) is highlighted with a square. The black circles indicate sites that provided complete mitochondrial genome sequences in previous studies (***Druzhkova et al., 2017***; ***Orlando et al., 2009***; ***Vilstrup et al., 2013***; ***Yuan***

*Figure 1 continued*

***et al., 2019***). The temporal range covered by the different samples analyzed is given in years before present (YBP) and follows the name of each site. Numbers between parentheses indicate the number of samples for which DNA sequence data could be generated. (**B**) *Facies masticatoria dentis* of P2, M3, p2, and m3 for the *E. (Susseminous) ovodovi* samples of the Honghe site analyzed here (**a**), *E. Sussemionus* (***Eisenmann, 2010***) (**b**), and *E. caballus* (Laboratory specimen) (**c**). 1, 4 protocones; 2, 5 metacones; 3 caballine notch. Teeth from the right side are shown, except for *E. Sussemionus*. The erupted teeth of the samples of the Honghe site appear to be smaller than those of the *E. Sussemionus* specimen.

The online version of this article includes the following figure supplement(s) for figure 1:

**Figure supplement 1.** Aerial view of the Honghe site.

**Figure supplement 2.** Archaeological material investigated in this study.

In this study, we have sequenced the complete nuclear genome of Sussemiones specimens. This allowed us to not only solve the phylogenetic placement of Sussemiones within the *Equus* evolutionary tree, but also to time their divergence relative to other non-caballine equids, as well as to reconstruct their demographic trajectory until their extinction during the mid-Holocene.

## Results

### Archaeological samples and sequencing data

All the equine specimens investigated in this study showed morphological and genetic signatures (short fragments of the mitochondrial hypervariable region) distinct from those of extant horses and donkeys (***Figure 1—figure supplement 2***). The morphological differences were especially marked in the second and third molars, which appeared to be smaller than in modern horses, and reminiscent of the third molars paracones and metacones observed in Sussemiones specimens (***Figure 1B***). Combined, these samples were radiocarbon dated to 3456–4460 calibrated years before the present (cal BP), including a mid-second millennium BCE date for the most recent sample, HH13H (3270 ± 30 uncal. BP, i.e., 3456–3616 cal BP) (***Supplementary file 1b***). They could, thus, represent some of the latest surviving Sussemiones individuals prior to their extinction.

We next aimed at genetically characterizing the taxonomic status of these specimens using high-throughput DNA sequencing technologies. We extracted ancient DNA from a total of 26 specimens and sequenced the whole nuclear genome at ~0.002–13.4 times coverage, including four samples from Honghe that provided 13.4×, 3.9×, 1.1×, and 1.0× nuclear genome (***Supplementary file 1a***). Comparison of the X chromosome and autosomal coverage revealed the presence of 15 male and 11 female individuals (***Supplementary file 1c***).

### Taxonomic status

To assess whether the sequenced specimens belonged to the same taxonomic group or comprised different species, we carried out a principal component analysis (PCA), including all the equine species sequenced at the genome level (depth of coverage ≥1×) (***Figure 2A***, ***Figure 2—figure supplements 1 and 2***). For this, we downloaded 11 previously-published equine genomes representing all extant species of equids and the extinct quagga zebra (***Huang et al., 2015***; ***Jónsson et al., 2014***; ***Kalbfleisch et al., 2018***; ***Orlando et al., 2013***; ***Renaud et al., 2018***; ***Supplementary file 1d***). All the Chinese specimens analyzed in this study were found to cluster together along the first two PCA components, in a group that was distinct from all other equine species (***Figure 2A***, ***Figure 2—figure supplement 1***) but closer to non-caballine equine species than to the horse (***Figure 2A***). This suggested that they were all members of a unique taxonomic group, most related to non-caballine equids.

Maximum likelihood (ML) and Bayesian phylogenetic analyses including the nearly complete 17 mitochondrial genomes reported in this study (***Supplementary file 1a***, depth of coverage above 1×) confirmed their clustering with non-caballine equids, within a single monophyletic group that also included five previously characterized Sussemiones specimens (***Figure 2B***, ***Figure 2—figure supplements 3 and 4***, ***Supplementary file 1e***). This grouping was supported with maximal (100%) bootstrap values. This, and the PCA clustering, indicated that the different excavation sites investigated in this study in fact all provided specimens that belonged to the *E. (Sussemionus) ovodovi* species.

We also used complete mitogenomes to assess the diversity of maternal lineages present in the *E. (Sussemionus) ovodovi* lineage. Phylogenetic analyses showed two major clades, of which only one

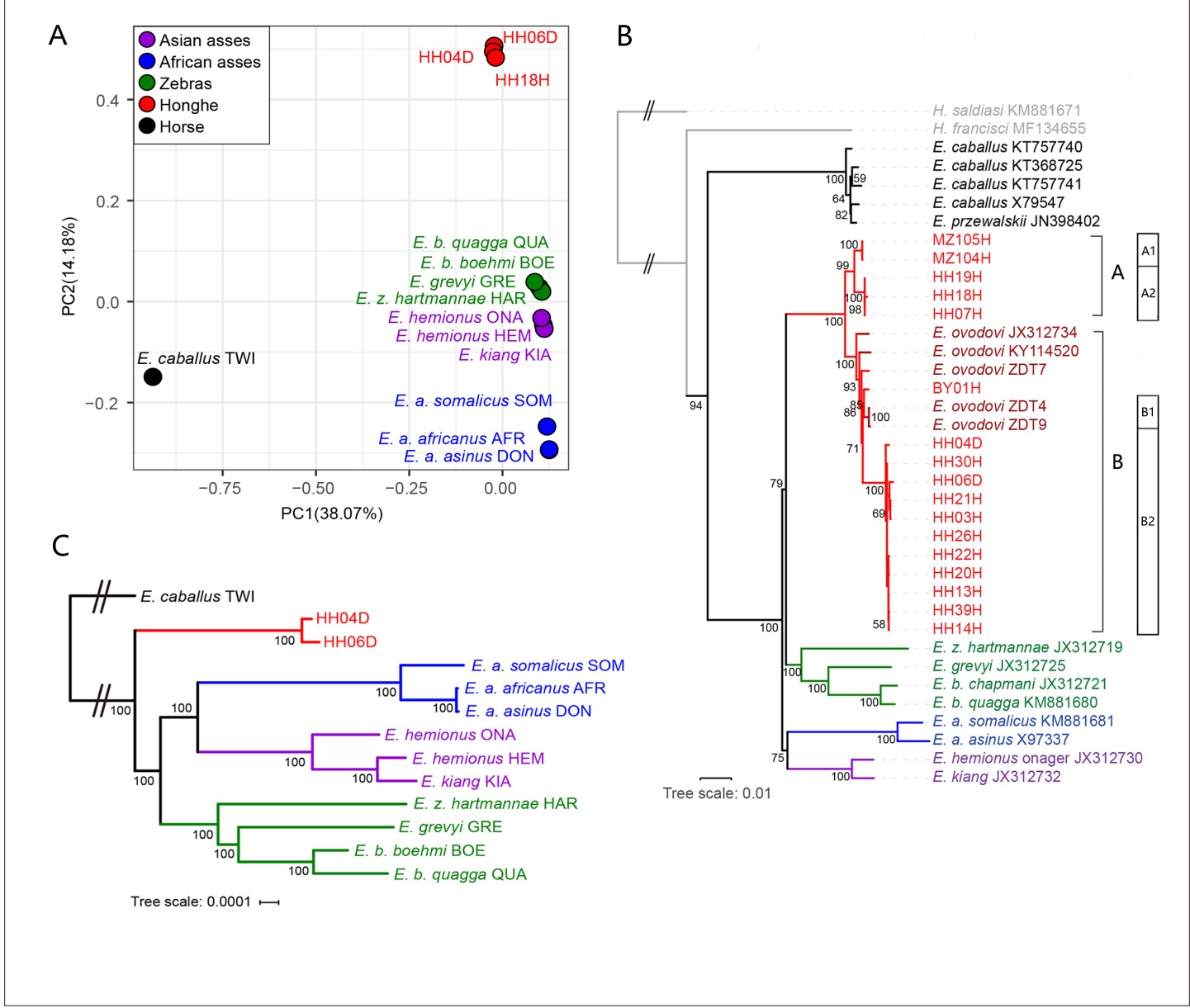

**Figure 2.** Genetic affinities within the genus *Equus*. The Honghe (HH), Muzhuzhuliang (MZ), and Shatangbeiyuan (BY) specimens are shown in red, while Asian asses, African asses, zebras, and horses are shown in purple, blue, green, and black, respectively. (**A**) Principal component analysis (PCA) based on genotype likelihoods, including horses and all other extant non-caballine lineages (16,293,825 bp, excluding transitions). Only specimens whose genomes were sequenced at least to 1.0× average depth of coverage are included. (**B**) Maximum likelihood tree based on six mitochondrial partitions (representing a total of 16,591 bp). Those *E. ovodovi* sequences that were previously published are shown in red. The tree was rooted using *Hippidion saldiasi* and *Haringtonhippus francisci* as outgroups. Node supports were estimated from 1000 bootstrap pseudo-replicates and are displayed only if greater than 50%. The black line indicates the mitochondrial clades A and B. (**C**) Maximum likelihood tree based on sequences of 19,650 protein-coding genes, considering specimens sequenced at least at a 3.0× average depth of coverage (representing 32,756,854 bp).

The online version of this article includes the following figure supplement(s) for figure 2:

**Figure supplement 1.** Principal component analysis (PCA) based on genotype likelihoods using the horse reference genome.

**Figure supplement 2.** Principal component analysis (PCA) based on genotype likelihoods using the donkey reference genome.

**Figure supplement 3.** RAxML-NG (GTR+GAMMA model) maximum likelihood phylogeny of complete mitochondrial sequence data.

**Figure supplement 4.** Bayesian mitochondrial phylogeny based on six partitions and using *Hippidion saldiasi* as outgroup.

**Figure supplement 5.** Exome-based maximum likelihood phylogeny rooted by the horse lineage.

**Figure supplement 6.** TreeMix analysis based on transversions and using the horse reference genome.

*Figure 2 continued on next page*

*Figure 2 continued*

**Figure supplement 7.** TreeMix analysis based on transversions and using the donkey reference genome.

**Figure supplement 8.** DNA damage patterns and the mapped read length distribution plots from mapDamage2 for HH06D.

**Figure supplement 9.** Error profiles of the 26 ancient genomes characterized in this study.

(clade B) was previously characterized. The other clade (A) consisted of two and three individuals from Muzhuzhuliang and Honghe, respectively (*Figure 2B*).

To further assess phylogenetic affinities, we used the two genomes characterized to at least 3× average depth of coverage (HH04D and HH06D) to place Sussemiones within the equine phylogenetic tree. To achieve this, we used ML phylogenetic reconstruction and an alignment of the coding sequences of the protein-coding genes (*Figure 2C*, *Figure 2—figure supplement 5*). This showed that the Chinese ancient specimens branched off before the radiation leading to modern asses and zebras (*Figure 2C*). Similar tree topologies were recovered using whole-genome SNPs by TreeMix (*Pickrell and Pritchard, 2012*; *Figure 2—figure supplements 6 and 7*, *Supplementary file 1f*). Combined with the analysis of the occlusal surface of the molars, in particular the absence of the caballine notch, the shape of metacones and protocones, and the reduced tooth size (*Figure 1B*), our analyses consistently supported the material analyzed as small specimens of the extinct *Equus* (*Sussemionus*) *ovodovi*. We, thus, concluded that this lineage survived in China during the Holocene, and until 3477–3637 cal BP, which is ~9000 years after the latest known specimen to date (*Druzhkova et al., 2017*; *Orlando et al., 2009*; *Vilstrup et al., 2013*; *Yuan et al., 2019*).

## Interspecies admixture and demographic modeling

Bifurcating trees fail to capture possible admixture events between lineages. Yet, previous research has unveiled pervasive admixture within equids, even amongst extant equids showing different chromosomal numbers (*Jónsson et al., 2014*). We thus next assessed whether the genomic data showed evidence for gene flow between Sussemiones and other non-caballine equids. To achieve this, we first applied D-statistics (*Soraggi et al., 2018*) to the genome sequence underlying 26 individual genomes and detected that *E. ovodovi* shared an excess of derived polymorphisms with asses than relative to zebras (*Figure 3—figure supplements 1 and 2*). This suggested that at least one admixture event could have taken place between Sussemiones and the ancestor of asses after their divergence from zebras.

We next leveraged the comparative genome panel and the ancient *E. (Sussemionus) ovodovi* genome characterized to high depth of coverage (HH06D) to reconstruct the equine demographic history using G-PhoCS (*Gronau et al., 2011*). More specifically, we first selected members of each equine lineage representing a total number of 10 genomes, conditioned analyses on 15,324 'neutral' loci, and assumed that the genus *Equus* emerged some 4.0–4.5 Mya, following previous estimates (*Orlando et al., 2013*). G-PhoCS analyses confirmed previous work indicating that the zebras and asses linages diverged ~2.0 Mya and that the deepest divergence within zebras and asses took place prior to ~1.5 Mya (*Jónsson et al., 2014*; *Figure 3*). It revealed that the Sussemiones lineage diverged from the ancestors of extant non-caballine equids ~2.3–2.7 Mya, in line with the fossil record (*Eisenmann, 2010*). Allowing for migrations provided support for gene flow between Sussemiones and the ancestor of asses and zebras (*Figure 3*). However, weak to no migrations were detected between Sussemiones and extant equids (*Supplementary file 1g*). Importantly, the admixture between Sussemiones and the ancestor of extant asses seems to have been stronger than that between Sussemiones and the ancestor of extant zebras, in line with the results of D-statistics. G-PhoCS also supported the presence of significant unidirectional gene flow prior to ~2.3–2.7 Mya, from the horse branch into the ancestral branch to all non-caballine equids, including Sussemiones (probability of gene flow 2.2–8.8%, *Supplementary file 1h*). This is consistent with previous HMMCoal analyses applied to whole-genome sequences of all extant equine species, which indicated significant gene flow between the deepest branches of the *Equus* phylogenetic tree until 3.4 Mya, mostly from a caballine lineage into the ancestor of all non-caballine equids (*Jónsson et al., 2014*).

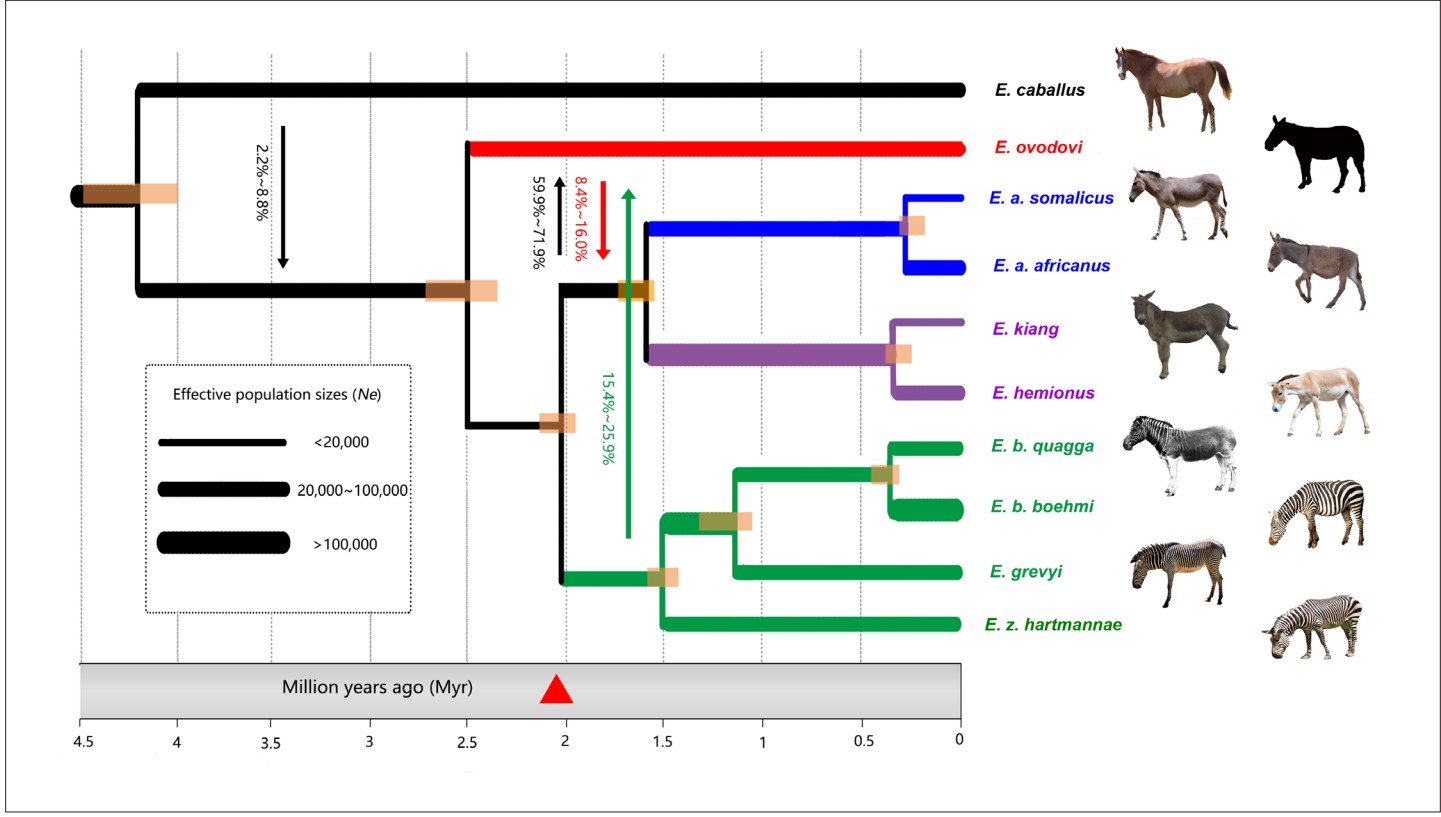

**Figure 3.** Demographic model for extinct and extant equine lineages as inferred by G-PhoCS (*Gronau et al., 2011*). Node bars represent 95% confidence intervals. The width of each branch is scaled with respect to effective population sizes ($N_e$). Independent $N_e$ values were estimated for each individual branch of the tree, assuming constant effective sizes through time. Migration bands and probabilities of migration (transformed from total migration rates) are indicated with solid arrows. The red triangle indicates the earliest *Sussemionus* evidence found in the fossil record. (Images: *E. caballus* by Infomastern, *E. a. somalicus* by cuatrok77, *E. kiang* by Dunnock_D, *E. a. africanus* by Jay Galvin, *E. hemionus* by Cloudtail the Snow Leopard, *E. z. hartmannae* by calestyo, *E. b. quagga* by Internet Archive Book Images, *E. b. boehmi* by GRIDArendal, and *E. grevyi* by 5of7.)

The online version of this article includes the following figure supplement(s) for figure 3:

**Figure supplement 1.** D-statistics in the form of (zebra, ass; *E. ovodovi*, outgroup), using sequence alignments against the horse reference genome.

**Figure supplement 2.** D-statistics in the form of (zebra, ass; *E. ovodovi*, outgroup), using sequence alignments against the donkey reference genome.

**Figure supplement 3.** Neighbor-joining (NJ) tree of selected samples based on 15,324 candidate 'neutral' loci identified using sequence alignments against the horse reference genome (detailed in 'Data preparation and filtering').

## Dynamic demographic profiles, heterozygosity, and inbreeding levels

We next leveraged the high-coverage Sussemiones genome characterized here to further explore the demographic dynamics until extinction. When modeled as constant through time, population sizes in G-PhoCS indicated that most lineages, including Sussemiones, consisted of small populations, excepting the Burchell's zebra (*Supplementary file 1i*). Pairwise sequential Markovian coalescent (PSMC) analyses, however, provided evidence for population size variation through time. First, the PSMC demographic trajectory of Sussemiones was found to diverge from that of other non-caballine equids (specifically, *E. hemionus*) after ~2.0 Mya, confirming the divergence date estimate retrieved by G-PhoCS (*Supplementary file 1i*). Second, the Sussemiones demographic trajectory was found to have constantly increased during the last million year but to have remained relatively low for a long period of time, until it reached a peak between 74 and 84 kya. It was, then, followed by an ~45-fold collapse until 13 kya (*Figure 4*). The lineage maintained extremely reduced population sizes through the Last Glacial Maximal (LGM, 19–26 kya) (*Clark et al., 2009*) and the Holocene, until it ultimately became extinct.

Importantly, the sample sequenced to sufficient coverage (HH06D) showed minimal heterozygosity and moderate inbreeding levels identified by the fraction of the segments within runs of homozygosity

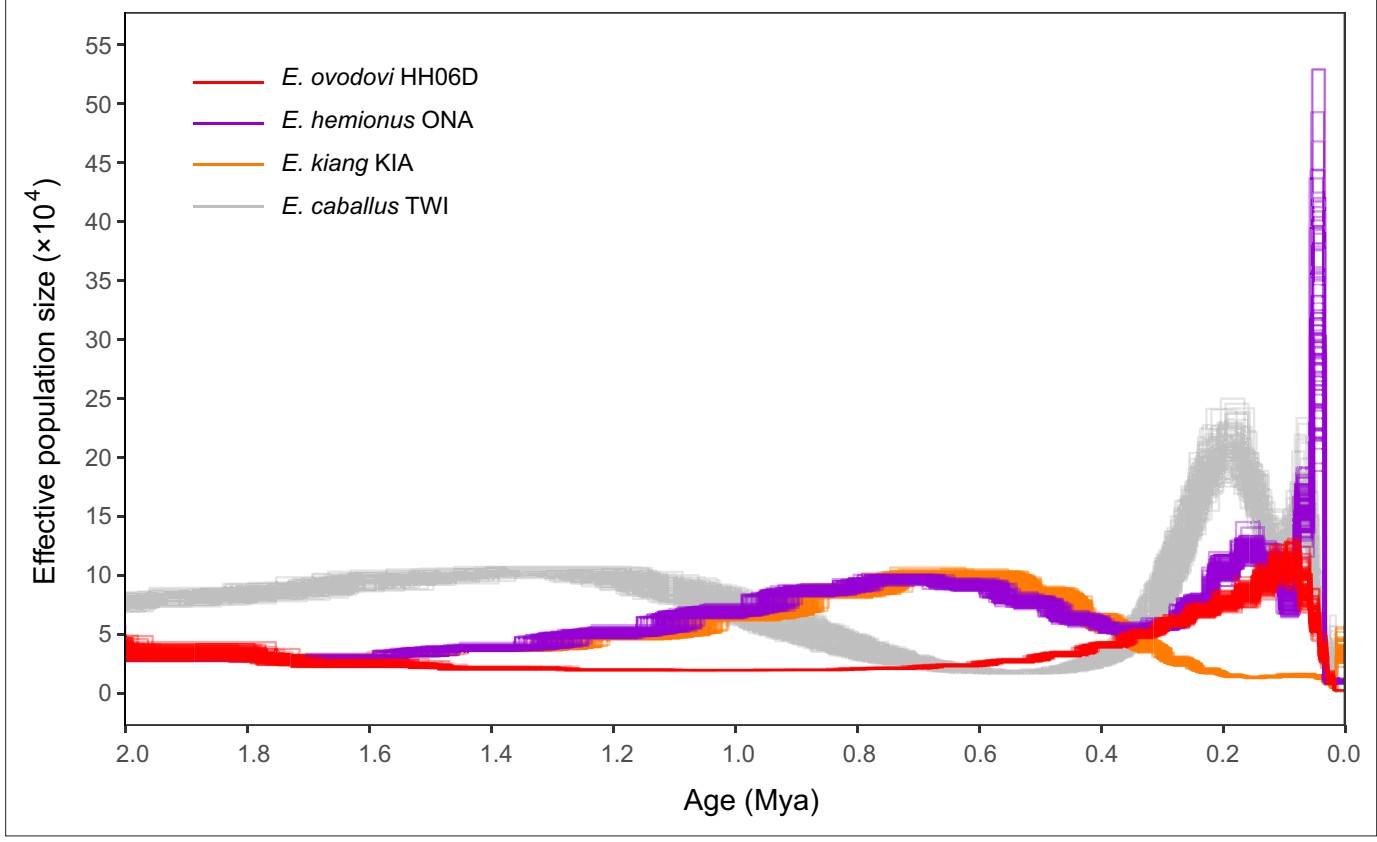

**Figure 4.** Pairwise sequential Markovian coalescent (PSMC) profiles (100 bootstrap pseudo-replicates) of four Eurasian equine species (*E. ovodovi* HH06D, *E. caballus* TWI [***Kalbfleisch et al., 2018***], *E. hemionus* ONA, and *E. kiang* KIA) (***Jónsson et al., 2014***). The y-axis represents the effective population size (×10,000), and the x-axis is scaled in millions of years before present. Faded lines show bootstrap values.

The online version of this article includes the following figure supplement(s) for figure 4:

**Figure supplement 1.** Pairwise sequential Markovian coalescent (PSMC) bootstrap pseudo-replicates for samples with (left) and without (right) transitions.

**Figure supplement 2.** Determining the uniform false-negative rate (uFNR) that was necessary for scaling pairwise sequential Markovian coalescent (PSMC) results.

(ROH) (*Figure 5*). Strikingly, this is true in spite of the increased DNA damage error rates of this genome (*Figure 2—figure supplement 9*), which likely inflate our estimates. The limited population sizes and resulting genetic diversity, rather than particularly enhanced inbreeding, may, thus, have limited the chances of survival of the species and have ultimately led to extinction.

## Discussion

### Phylogenetic placement of *Equus* (*Sussemionus*) *ovodovi*

In this study, we have characterized the first nuclear genomes of the now-extinct equine lineage, *E.* (*Sussemionus*) *ovodovi*, the last surviving member of the subgenus *Sussemionus*. We demonstrated that this lineage survived in China well into the Holocene with the most recent specimens analyzed dating to ~3456–3616 cal BP. This is almost 9000 years after the latest specimens previously documented in the fossil record (***Druzhkova et al., 2017***; ***Vilstrup et al., 2013***; ***Yuan et al., 2019***). Our work, thus, shows that *Sussemionus* represents the last currently known *Equus* subgenus to become extinct. Our work also adds to the list of recently identified members of the horse family that were still alive at the time horses and donkeys were first domesticated, ~5500 years ago (***Fages et al., 2019***; ***Gaunitz et al., 2018***; ***Rossel et al., 2008***). In contrast to those divergent members that were identified in Siberia (*Equus lenensis*) and Iberia (IBE), which both belonged to the horse species (***Fages et al., 2019***; ***Schubert et al., 2014a***), Sussemiones members were most closely related to

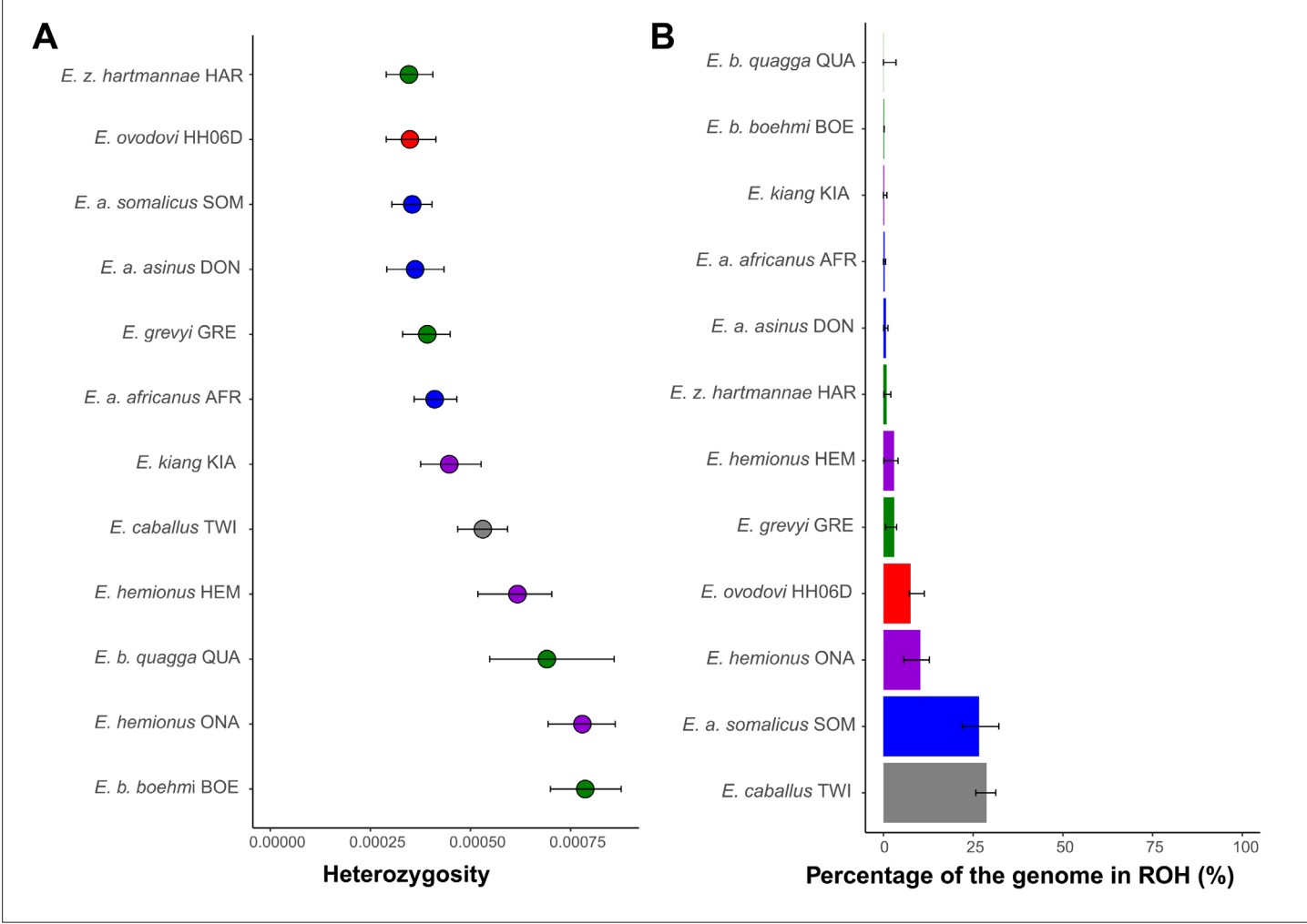

**Figure 5.** Heterozygosity and inbreeding levels of different equine lineages. (**A**) Individual heterozygosity outside runs of homozygosity (ROH). (**B**) Fraction of the genome in ROH. Estimates were obtained excluding transitions and are shown together with their 95% confidence intervals. The colors mirror those from *Figure 2*.

The online version of this article includes the following figure supplement(s) for figure 5:

**Figure supplement 1.** Heterozygosity rates outside runs of homozygosity (ROH) together with 95% confidence intervals.

**Figure supplement 2.** The fraction of the genome segments consisting of runs of homozygosity (ROHs) together with 95% confidence intervals.

non-caballine equids. This is in agreement with previous studies (*Der Sarkissian et al., 2015*; *Druzhkova et al., 2017*; *Heintzman et al., 2017*; *Orlando et al., 2009*; *Vilstrup et al., 2013*; *Yuan et al., 2019*), which could, however, not fully resolve the exact phylogenetic placement of this species within non-caballines as topological tests based on mitochondrial genomes received low confidence support (*Der Sarkissian et al., 2015*; *Druzhkova et al., 2017*; *Heintzman et al., 2017*; *Orlando et al., 2009*; *Vilstrup et al., 2013*; *Yuan et al., 2019*). Our study solved this question by reporting the first whole-genome phylogeny of Sussemiones, which confirmed with maximal bootstrap support this species as a basal lineage of non-caballine equids.

## Suitable habitat and geographic distribution

Previous zooarchaeological and environmental research indicated an ecological range for Sussemiones overlapping with the grasslands located east of the Altay Mountains and west of the Yenisei River during the Late Pleistocene (*Khenzykhenova et al., 2016*; *Malikov, 2016*; *Plasteeva et al., 2015*; *Shchetnikov et al., 2015*; *Slon et al., 2018*). Recent research also reported species occurrence in northeastern China (~12,600–40,200 YBP), where similar climatic and ecological conditions

were found at the time (*Yuan et al., 2019*). It could, thus, be speculated that Sussemiones were adapted to an environment with moderately dry climatic conditions and steppe landscapes (*Yuan et al., 2019*). However, our study identified Sussemiones specimens in three late Holocene sites from China characterized by mild and humid environmental conditions. Additionally, two distinct mitochondrial haplogroups from 22 individuals have been defined from the six known sites, suggesting possible population structure across various geographic areas and adaptation to local environments. It also suggests that the species could adapt to a wider variety of habitats than previously hypothesized and rejects the contention that the species became extinct as it could not survive in warmer climatic conditions (*Yuan et al., 2019*).

Interestingly, the Sussemiones specimens identified in this study were excavated from sites in northeastern China located at almost the same latitude as those Sussemiones localities known so far from Russia, but also at lower latitudes (*Figure 1A*). This implies that the geographic range of *E. ovodovi* was larger than previously expected and included at least Northern China and Southern Siberia. Although in the absence of identified fossils from Mongolia, given that there is a lack of mitochondrial phylogeographic structure, we could speculate that the two regions were in contact at least maternally. Further work is necessary to establish whether or not the species survived in other pockets both within and outside China.

## Demographic history with ancestral interspecific admixture

Our analyses reveal that the divergence between Sussemiones and the most recent common ancestor of all extant non-caballine equids took place ~2.3–2.7 Mya, prior to the divergence of zebras and asses. Post-divergence admixture events with the lineage ancestral to asses and zebras, on the one hand, and the lineage ancestral to all extant zebras, were also identified (*Figure 3* and *Supplementary file 1h*). Our results, thus, reveal non-caballine ancestral lineages occupying partly sympatric distributions that were, consequently, different than those of their descendants, in which zebras are restricted in Africa and Asian asses in Asia. Whether the admixture events identified here directly involved the Sussemiones lineage or one (or more) ghost lineage(s) closely related to Sussemiones requires further research.

## Limited genetic diversity before extinction

The demographic profile of Sussemiones shows that after the peak of population size culminating ~74 kya, Sussemiones went through a slow and continuous decline until 13 kya (*Figure 4*). This time period encompasses several major climate changes (especially the LGM, ~19–26 kya) (*Clark et al., 2009*) and the great human expansion to Eurasia (~35–45 kya) (*Henn et al., 2012*). The effective size of Sussemiones populations that survived in Northern China until at least ~3500 years ago, remained extremely small, as indicated by their extremely reduced heterozygosity levels compared to other extant and extinct equine species. As the inbreeding levels were not particularly high compared to some members of endangered equine species (*Figure 5*), the reduced genetic diversity available in the lineage may have compromised the long-term survival of the lineage, in a process partly reminiscent of what was previously described for the woolly mammoth (*Palkopoulou et al., 2015*). And considering the rapid expansion of domestic horses across Eurasia from about 2000 BC (*Librado et al., 2021*), this lineage was ultimately replaced under growing anthropogenic stress.

In conclusion, our study clarifies the phylogenetic placement, speciation timing, and evolutionary history of the now-extinct *Sussemionus* equine subgenus. This group did not remain in reproductive isolation from other equine lineages, but contributed to the genetic makeup of the ancestors of present-day asses, while receiving genetic material from the ancestors of African zebras. This supports geographic distributions at least partly overlapping at the time, thus, not identical to those observed today. The species demographic trajectory experienced a steady decline from ~74 kya and during a period witnessing both important climatic changes and the Great human expansion across Asia (*Henn et al., 2012*). It survived with minimal genetic diversity the Pleistocene-Holocene transition, and for at least eight millennia before it became extinct, which provides insights into the survival potential of large animals since the Holocene. Given the persistence of Sussemiones throughout the third and second millennia BCE, archaeologists must be exceedingly careful while assigning Asian zooarchaeological material to equine taxa until this period.

## Materials and methods

### Genome sequencing

Minimum number of individuals (MNI) was determined by assigning the frequency of hip bone and was calculated from the acetabular bone to avoid double counting. MNI was estimated to 31 individuals at Honghe, 4 at Muzhuzhuliang, and 4 at Shatangbeiyuan. DNA preservation conditions were compatible with the recovery of ancient DNA sequences from only 20 of the 31 Honghe samples, 3 of the 4 Muzhuzhuliang samples, and 3 of the 4 Shatangbeiyuan samples (*Supplementary file 1a*).

All pre-PCR procedures were conducted in a dedicated ancient DNA laboratory at Jilin University (JLU) that is physically separated from the post-PCR laboratory. To remove potential contaminant DNA, working areas and benches were frequently cleaned with bleach and UV exposure. Lab experiments were carried out wearing full-body suits, facemasks, and gloves. To detect contamination, mock controls were included at each experimental step, including DNA extraction, DNA library preparation, and PCR setup.

Prior to DNA extraction, the outer surface of the sample was cleaned with a brush. The cleaned sample was subsequently cut into smaller pieces and soaked in 10% bleach for 20 min, rinsed with ethanol and distilled water, and then UV-irradiated for 30 min on each side. Finally, powder was obtained using a dental drill (Traus 204, Korea). Ancient DNA was extracted from the sample powder by using a modified silica spin column method (*Yang et al., 1998*), in the dedicated ancient DNA facilities from JLU. For each specimen, a total of 200 mg powder was added with 3.9 ml EDTA (0.465 mol/L) and placed in the refrigerator at 4°C for 12 hr for decalcification, and then 0.1 mL proteinase K (0.4 mg/mL) were added and incubated overnight in a rotating hybridization oven at 50°C (220 rpm). After centrifugation, the supernatant was transferred into an Amicon Ultra-4 centrifugal filter device (Merck Millipore Ltd, 10,000 Nominal Molecular Weight Limit), reduced to less than 100 µL, and purified with QIAquick PCR Purification Kit (QIAGEN), according to the manual instructions.

Before preparation of DNA libraries, we first PCR-targeted short fragments of the mitochondrial hypervariable region to select those samples positive for the presence of equine DNA (which was further confirmed through Sanger sequencing). For this, we used the oligonucleotide primers L15473 5'-CTTCCCCTAAACGACAACAA-3' and reverse primer H15692 5'-TTTGACTTGGATGGGGTATG-3'; and forward primer L15571 5'-AATGGCCTATGTACGTCGTG-3' and reverse primer H15772 5'-GGGAGGGTTGCTGATTTC-3' from *Juan et al., 2007*, and the amplification conditions therein.

Double-stranded single-indexed libraries were prepared using NEBNext Ultra II DNA Library Prep Kit for Illumina (NEB #E7645S) and NEBNext Multiplex Oligos for Illumina Index Primers Set 1 and 2 (NEB #E7335S, #E7500S), following the manufacturer's instructions with minor modifications. Specifically, the extracted DNA (50 µL) were end-repaired and A-tailed by adding 7 µL of NEBNext Ultra II End Prep Reaction Buffer and 3 µl of NEBNext Ultra II End Prep Enzyme Mix, and incubated for 40 min at 20°C and then 30 min at 65°C. The adaptor was ligated to the dA-tailed DNA fragments by adding 30 µL of NEBNext Ultra II Ligation Master Mix, 1 µL of NEBNext Ligation Enhancer and 2.5 µL of NEBNext Adaptor for Illumina (dilution 1:10), and incubated for 20 min at 20°C. The adaptor was then linearized by adding 3 µL of USER Enzyme and performing an incubation for 15 min at 37°C. The adaptor-ligated DNA were cleaned without size selection using the MinElute PCR Purification Kit (QIAGEN, Germany), following the instructions provided by the manufacturer. PCR enrichment was performed by using 30 µL of NEBNext Ultra II Q5 Master Mix, 1 µL of Index Primer, 1 µL of Universal PCR Primer, and 18 µL of adaptor-ligated DNA. PCR cycling conditions comprised an initial denaturation at 98°C for 30 s, 14–16 cycles of 98°C for 10 s, 65°C for 75 s, and a final extension at 65°C for 5 min. PCR-amplified DNA libraries were purified using Agencourt AMPure XP Beads, following the manufacturer's instructions, and Illumina sequencing was performed on HiSeq X Ten platform using 150 bp paired-end reads. Overall, we sequenced a total of 28 DNA libraries and generated 2,727,843,803 read pairs (https://www.ebi.ac.uk/ena/browser/view/PRJEB44527?show=reads).

### Radiocarbon dating

Radiocarbon dating of the samples was performed at the Beta Analytic Radiocarbon Dating Laboratory, Miami, FL. Bone or tooth pieces about 2 g were sampled in the bone and sent for subsequent dating of collagen (not ultrafiltered). Calibration was carried out using OxCalOnline (https://c14.arch.ox.ac.uk/oxcal.html) and the IntCal20 calibration curve. Calibrated dates are provided in *Supplementary file 1b*.

## Data processing

Sequencing reads were processed and aligned against the horse (EquCab3.0 *Kalbfleisch et al., 2018*) and donkey (*Renaud et al., 2018*) reference genomes using the PALEOMIX pipeline (*Schubert et al., 2014b*) with default parameters, except that we followed the recommendations from *Schubert et al., 2012* and disabled seeding. Briefly, paired-end (PE) reads longer than 25 nucleotides were trimmed with AdapterRemoval v2.2 (*Schubert et al., 2016*) and aligned against the reference genomes using BWA (*Li and Durbin, 2009*), retaining alignments with mapping qualities superior to 25. PCR duplicates were then removed using Picard (http://broadinstitute.github.io/picard/) (*Broad Institute, 2019*). Finally, all ancient and modern reads were locally realigned around indels using GATK (*McKenna et al., 2010*).

Postmortem DNA damage and average sequencing error rates were determined with mapDamage2.0 (*Jónsson et al., 2013*; *Figure 2—figure supplement 8*) and ANGSD (*Korneliussen et al., 2014*; *Figure 2—figure supplement 9*), respectively. Further rescaling and trimming procedures were implemented following *Gaunitz et al., 2018* to limit the impact of remnant nucleotide misincorporations in subsequent analyses. For each of the DNA libraries examined, the base composition of the position preceding read starts on the horse reference genome showed an excess of Guanine and, to a lesser extent, of Adenine residues (*Figure 2—figure supplement 8*). This is in line with depurination driving postmortem DNA fragmentation (*Briggs et al., 2007*). Additionally, error rate estimates for each nucleotide substitution class indicated the predominance of C→T and G→A misincorporations (*Figure 2—figure supplement 9*). Such misincorporation rates were particularly inflated towards read ends, but not read starts (*Figure 2—figure supplement 8*). This is in line with the DNA nucleotide misincorporation profiles expected for the type of DNA library constructed (*Seguin-Orlando et al., 2015*), which was caused by the Q5 polymerase being unable to read through 5' uracils, thereby excluding the typical 5' excess of C-to-T. MapDamage profiles were, thus, consistent with Cytosine deamination at 5'-overhanging ends as the most prominent postmortem DNA degradation reactions (*Jónsson et al., 2013*).

GATK HaplotypeCaller was used to obtain individual gvcf files with "`--minPruning 1 --minDanglingBranchLength 1`" to increase sensitivity. Then, we employed GATK GenotypeGVCFs for genotyping with the option "`--includeNonVariantSites`" in order to retain non-variant loci. The vcf files were further filtered in TreeMix and G-PhoCS analysis.

## Principal component analysis (PCA)

The genotype likelihood framework implemented in ANGSD helped mitigate various error rates in ancient and modern genomes. Using EquCab3 (*Kalbfleisch et al., 2018*) as the reference genome, ANGSD was run using the following options: "-only_proper_pairs 1 -uniqueOnly 1 -remove_bads 1 -minQ 20 -minMapQ 25C 50 -baq 1 -skipTriallelic 1GL 2 -SNP_pval 1e-6 -rmTrans 1". This provided a dataset consisting of a total of 16,293,825 transversions when the horse was included, and 10,094,431 transversions when the horse was excluded (i.e., when analyses were restricted to non-caballine genomes only). In these analyses, only specimens sequenced to an average depth of coverage ≥1× were retained. PCA was carried out using the PCAngsd package (*Meisner and Albrechtsen, 2018*; *Figure 2A*). To assess the impact of potential reference bias, all analyses were repeated after mapping the sequence data against the donkey reference (*Figure 2—figure supplement 2*).

## Phylogenetic inference

### Mitochondrial phylogeny

Cleaned reads were mapped against the horse mitochondrial genome (GenBank accession no. NC_001640), following the same procedure as when mapping against the nuclear genome. Samples showing an average depth of coverage <1× were disregarded, leaving a total of 17 individuals for further analyses. After removing duplicates, consensus mitochondrial sequences were generated using ANGSD (-doFasta 2 -doCounts 1 -setMinDepth 3 -uniqueOnly 1 -remove_bads 1 -minQ 25 -minMapQ 25). Multiple alignment was performed together with the comparative mtDNA sequences downloaded from GenBank (*Supplementary file 1e*) using MUSCLE v3.8.31 (*Edgar, 2004*), with default parameters. The alignments were then split into six partitions (first, secoond, and third codon positions, rRNA, tRNA, and control region) by Partition Finder v2.1.1 (*Lanfear et al., 2012*).

Two ML trees based on all six partitions and excluding the control region (positions 15,469–16,660 of the horse reference mitochondrial genome) were both reconstructed using RAxML-NG v.0.9.0 (*Kozlov et al., 2019*) with GTR+GAMMA substitution model. A total of 1000 bootstrap pseudo-replicates were carried out to assess node robustness (*Figure 2—figure supplement 3*). BEAST 2.6.6 (*Bouckaert et al., 2019*) was used to perform Bayesian phylogenetic reconstruction and to estimate split times. The six partitions described above were used, for which the best substitution model was determined using modelgenerator (version 0.85, *Keane et al., 2006*) and a Bayesian information criterion. We calibrated the tree using tip dates (see *Supplementary file 1j*) and an age of 4–4.5 Mya for the root of crown group *E. caballus* (normal prior, mean 4.25 Mya, SD: 0.15 Mya) (*Orlando et al., 2013*). We applied together with the birth-death model and a relaxed molecular clock (log normal) for 1000 million generations (sampling frequency = 1 every 1000), while forced monophyly for all main lineages, including donkeys, hemiones, horses, *ovodovi,* and zebras. Convergence was assessed visually using Tracer v1.6 (with all individual ESS >200), and posterior date estimates were retrieved using 25% as burn-in. The final consensus tree was produced by TreeAnnotator 2.6.6 (*Drummond and Rambaut, 2007*) as the maximum clade credibility tree from 100,000 randomly sampled trees obtained using LogCombiner v2.6.6 (*Bouckaert et al., 2019*) (burn-in = 20%). The final tree was plotted using ITOL (*Letunic and Bork, 2016*; *Figure 2—figure supplement 4*).

## Autosomal phylogeny

As for autosomes, we reconstructed an ML phylogenetic tree as implemented in the PALEOMIX phylo_pipeline, which is dedicated to phylogenomic reconstructions (*Schubert et al., 2014b*). This analysis was based on the coding sequence (CDS) of protein-coding genes annotated in EquCab3.0, partitioning data according to first, second, and third codon positions. ML phylogenetic inference was performed using ExaML v3.0.21 (*Kozlov et al., 2015*) and RAxML v8.2.12 (*Stamatakis, 2014*) under the GAMMA substitution model with 100 bootstrap pseudo-replicates (*Figure 2C*, *Figure 2—figure supplement 5A*). We also repeated the same procedure after mapping against the donkey reference genome, which returned the same topology (*Figure 2—figure supplement 5B*).

Additionally, we extracted biallelic single-nucleotide polymorphisms (SNPs) from the dataset generated in the section 'Variant calling' using bcftools v1.9 (*Li et al., 2009*). Both variant datasets obtained following mapping against the horse and donkey reference genomes were used in this analysis to rule out reference bias. We applied filters composed of minimum Phred-scaled quality score quality (QUAL) = 20, sites for all individuals below 2 or twice the mean coverage, and allowed up to three individuals with missing data per site. After disregarding transitions, a total of 18,803,101 (mapping against horse genome) and 19,459,070 (mapping against donkey genome) transversions were finally used as input for TreeMix (*Pickrell and Pritchard, 2012*) with parameters "-k 500 -root TWI", and considering an increasing number of migrations edges (0 ≤ m ≤ 3; *Figure 2—figure supplements 6 and 7*, *Supplementary file 1f*).

## Admixture analyses with D-statistics

D-statistics were calculated to investigate potential introgression between *E. ovodovi* and other non-caballines (*Figure 3—figure supplement 1*) using the doAbbababa2 program in ANGSD (*Soraggi et al., 2018*). Individuals were grouped according to their respective species. D-statistics were computed in the form (((H1, H2), H3), Outgroup) considering only the autosomal sites from bam files mapping against the horse reference with the following options: "-minQ 20 -minMapQ 25 -remove_bads 1 -only_proper_pairs 0 -uniqueOnly 1 -baq 1C 50". The horse reference genome was used as the Outgroup. H1 and H2 denoted any non-caballine genomes except *E. ovodovi*, while H3 denoted the *E. ovodovi*. Confidence intervals were estimated applying a jackknife procedure and 5 Mb windows. Z-scores with absolute values higher than 3 were considered to be statistically significant. To rule out possible reference bias, we also rerun the same analysis using sequence alignments against the donkey reference genome (*Figure 3—figure supplement 2*).

## G-PhoCS demographic model

### Data preparation and filtering

In order to model the equine evolutionary history, we selected a total of 10 individuals representing each individual lineage and used their high-coverage genomes as input for G-PhoCS (*Gronau et al.,*

*2011*). Genotypes were called by GATK and candidate 'neutral' loci were identified by applying the following filters:

1. The simple repeats track available for the reference genome was obtained from Ensembl v99 release; corresponding regions were masked.
2. All exons of protein-coding genes were discarded together with their 10 kb flanking regions; this was done based on the GTF format annotation file of the reference genome available from Ensembl v99 Genome Browser.
3. We identified conserved noncoding elements (CNEs) using phastCons scores (based on the 20-way Conservation track provided for the mammal clade according to the genomic coordinates of the human reference) downloaded from the Table Browser of UCSC. All CNEs and their 100 bp flanking regions were masked using liftOver to convert human genome coordinates into EquCab3.0 horse genome coordinates.
4. Exons of noncoding RNA genes together with their 1 kb flanking regions were removed, based on the annotations available for the reference genome.
5. Gaps in the reference genome were disregarded.

Besides the various filters described above, regions/sites likely to be enriched for misaligned bases, and to have high false-negative rates during read alignment or variant detection, were masked as missing data. More specifically, different individuals could be treated differently depending on the genotyping results obtained as described in the section 'Variant calling,' depending on the presence of (1) indels, (2) triallelic sites, (3) positions with depth of coverage twice the mean depth recorded for each individual, and (4) transition sites.

We selected 1 kb loci located with minimum inter-locus distance of 30 kb from the intervals that pass all the criteria described above. Then, consensus sequences were generated for each individual from the vcf file generated in the section 'Variant calling' using bcftools 'consensus' command, with IUPAC codes indicating heterozygous genotypes (--iupac-codes) and 'N' representing masked sites (--mask and --missing 'N').

Finally, we excluded contiguous intervals if the total amount of missing bases was greater than 50% of the region length, resulting in a final collection of 15,324 loci using the horse reference genome (autosomes only). Neighbor-joining trees were constructed to confirm the topology before the inferring the population divergence (*Figure 3—figure supplement 3*).

## MCMC setup

We used default global settings (*Gronau et al., 2011*), including a gamma prior distribution ($\alpha$ = 1, $\beta$ = 10,000) for all mutation-scaled population sizes ($\theta$) and a gamma prior distribution ($\alpha$ = 0.002, $\beta$ = 0.00001) for all mutation-scaled migration rate ($m$). The initial parameter value of mutation-scaled divergence times ($\tau$) was first set individually for each population. Then, we ran ~100,000–200,000 iteration tests and manually evaluated the convergence by checking the achieve acceptance ratios (i.e., accept if around 30–70%) or using Tracer v1.6 (http://tree.bio.ed.ac.uk/software/tracer/). For each test, we updated the input of the initial $\tau$ and all fine-tuned parameters based on previous results to get the appropriate value. The final results in *Figure 3* are based on 500,000 MCMC iterations, considering the first 10% as burn-in.

## Parameter calibration

We assumed an average generation time ($g$) of 8 years. The coalescent time of the *Equus* (4.0–4.5 Mya) (*Orlando et al., 2013*) was used to bound the mutation rate $\mu$ (per site per year). Effective population sizes ($Ne$) and divergence times ($T$) were estimated by scaling $\theta$ and $\tau$ parameter using $g$ and $\mu$ (*Supplementary file 1g*), and the following formula: $Ne = \theta/(4 \mu g)$ and $T = \tau/\mu$ (*Gronau et al., 2011*).

## Inferring gene flow

Total migration rates ($M$) were estimated by a mutation-scaled parameter ($m$) given by $M = m\tau_m$, where $\tau_m$ is the mutation-scaled time span of the migration band. The total migration rate gives an accumulated migration rate over a long period of time, which can be superior to 100%. We then converted such rates, $M$, into a probability of migration using the formula $P = 1-e^{-M}$ (where $p$ is the probability of gene flow), according to the method presented in *vonHoldt et al., 2016*.

The migration model implemented in G-PhoCS makes it possible to detect gene flow between any two lineages by introducing migration bands manually to the demographic model. However, it remains difficult to detect weak migration events. Additionally, scenarios including a large number of migration bands can lead to spurious results. To address this, we first inferred a demographic model with no migration bands, and then introduced several migration bands corresponding to five independent scenarios (*Supplementary file 1g*). A significant migration band was considered supported if both the 95% Bayesian credible interval of total migration rate (*M*) did not include 0% values and if the mean value of *M* was estimated to be greater than 0.03%.

Settings for the migration bands between extant caballines are based on previous research (*Jónsson et al., 2014*). The significant migration band from horse to the non-caballine ancestor was identified (*Supplementary file 1g*), in line with previous work (*Jónsson et al., 2014*). However, no other non-negligible (*M* > 3%) migration bands were found in our analyses (*Supplementary file 1g*).

We then tried to estimate the migration events between *E. ovodovi* and other branches. We added all possible migration bands between *E. ovodovi* and extant non-caballine branches into the demographic model except the migration bands between *E. ovodovi* and the ancestor of extant non-caballines as the model is often underpowered to infer migration between sister populations. All of the migration bands were separated into four demographic models. Only three migration bands were shown to be significant (*Supplementary file 1g*).

Finally, the total four migration bands were combined into one demographic model (*Supplementary file 1h*) and compared the estimates to the one including no migration (*Supplementary file 1i*).

We caution that the analyses carried out using TreeMix and G-PhoCS returned partly discordant results. This may be due to TreeMix modeling pulses of admixture in contrast to G-PhoCS, in which situations of continuous gene flow can be accommodated. Additionally, gene flow affecting the two deepest tree branches can be directly accommodated by reducing their divergence. Therefore, the deep admixture inferred by G-PhoCS from the caballine branch into the ancestral branch of Sussemiones and other non-caballine equids cannot be expected to be identified through an individual migration edge with TreeMix as this could simply be modeled through a more limited divergence between both underlying lineages. The same holds true for the asymmetric gene flow inferred by G-PhoCS between Sussemiones and the branch ancestral to all extant asses; TreeMix is likely to only identify the resulting unidirectional contribution of these admixtures, which mainly sources to the branch ancestral to extant asses into Sussemiones; since G-PhoCS also infers additional admixture from the branch ancestral to extant zebras into Sussemiones, we can expect TreeMix to accommodate both sources of gene flow through a reduced divergence between Sussemiones and the branch ancestral to stenonines. Finally, it may reflect limitations pertaining to the two underlying data sets utilized, consisting, on the one hand, to the whole-genome SNP panel for TreeMix, further filtered for 15,324 candidate 'neutral' loci in G-PhoCS.

## Demographic trajectories with PSMC
### PSMC analyses
In order to reconstruct the past demographic dynamics of the *E. ovodovi* lineage, we applied the PSMC algorithm (version 0.6.5-r67) (*Li and Durbin, 2011*) to the sample HH06D (12.0×, mapping against horse reference), as well as three other Eurasian equine species (*E. caballus* TWI, *E. hemionus* ONA, and *E. kiang* KIA).

We first obtained the diploid consensus sequences after mapping against the horse genome for the autosomes of each specimens using bcftools 'mpileup' command and the 'vcf2fq' command from vcfutils.pl with the following filters: mapping quality ≥ 25; adjust mapping quality = 50; minimum depth of coverage = 8; maximum depth of coverage ≤ 99.5% quantile of the coverage distribution; minimum RMS mapping quality = 10; filtering window size of indels = 5.

After filtering the bases with Phred quality scores strictly lower than 35, we ran PSMC with the following command: 'psmc -N25 -t15 -r5 -p "4+25*2+4 + 6"'. Calibration was carried out using a generation time of 8 years and mutation rate of $7.242 \times 10^{-9}$ per generation per site, following previous work (*Jónsson et al., 2014*). However, as for the misincorporation pattern and high error rate of HH06D (*Figure 2—figure supplements 8 and 9*), we also performed analyses without transitions using mutation rates of $2.3728 \times 10^{-9}$ that was obtained assuming that the most recent common ancestor of living equine species emerged 4 Mya (*Orlando et al., 2013*).

We found a great expansion of HH06D in the past 50,000 years when retaining transitions but not when conditioning on transversions (*Figure 4—figure supplement 1*). The former is thus likely spurious and at least partly driven by severe postmortem DNA damage signatures in the sequence data. We therefore only used the latter inference when considering the ancient HH06D specimen.

## False-negative rate correction

The HH06D genome (12.0×) was corrected assuming a uniform false-negative rate (uFNR) following *Orlando et al., 2013* as the average depth of coverage is lower than the recommended 20×. To identify the correction value of uFNR for HH06D, we randomly downsampled reads of the SOM genome (21.0×), using DownsampleSam function of Picard Tools to downscale sequence data to the same average depth of coverage as that obtained for HH06D. This indicated that a value of 0.22 was the most suitable uFNR value for rescaling the HH06D PSMC profile (*Figure 4—figure supplement 2A*). The KIA and the ONA genomes, which also showed limited coverage, were also rescaled following the same procedure (*Figure 4—figure supplement 2B and C*). Finally, PSMC confidence intervals were assessed from 100 bootstrap pseudo-replicates (*Figure 4*).

## Heterozygosity inference and inbreeding

Global heterozygosity rates and inbreeding levels were inferred for high-coverage individuals (>10×) using ROHan (*Renaud et al., 2019*) with default parameters, except that transitions were excluded (--tvonly) (*Figure 5—figure supplement 1*). To limit the impact of remnant misincorporations, we used the attached estimateDamage.pl script to estimate damage for all ancient samples prior to heterozygosity computation. Inbreeding was co-estimated together with genome-wide heterozygosity levels from the total ROH length (*Figure 5—figure supplement 2*).

# Acknowledgements

We thank High-Performance Computing (HPC) of Northwest A&F University (NWAFU) for providing computing resources.

# Additional information

### Funding

| Funder | Grant reference number | Author |
| --- | --- | --- |
| Major Program of National Fund of Philosophy and Social Science of China | 17ZDA221 | Dawei Cai |
| H2020 European Research Council | 681605 | Ludovic Orlando |
| National Natural Science Foundation of China | 31822052 | Yu Jiang |

The funders had no role in study design, data collection and interpretation, or the decision to submit the work for publication.

### Author contributions

Dawei Cai, Conceptualization, Data curation, Formal analysis, Funding acquisition, Investigation, Methodology, Project administration, Resources, Supervision, Validation, Writing - original draft, Writing - review and editing; Siqi Zhu, Mian Gong, Conceptualization, Data curation, Formal analysis, Investigation, Methodology, Project administration, Software, Validation, Visualization, Writing - original draft, Writing - review and editing; Naifan Zhang, Weilu Sun, Xinyue Shao, Yaqi Guo, Investigation, Validation; Jia Wen, Data curation, Formal analysis, Investigation, Software, Validation, Visualization; Qiyao Liang, Investigation, Validation, Visualization; Yudong Cai, Zhuqing Zheng, Data curation, Investigation, Methodology, Software; Wei Zhang, Songmei Hu, Xiaoyang Wang, He Tian, Youqian Li, Wei Liu, Miaomiao Yang, Jian Yang, Duo Wu, Resources; Ludovic Orlando, Conceptualization,

Methodology, Supervision, Writing - review and editing; Yu Jiang, Conceptualization, Funding acquisition, Methodology, Project administration, Resources, Supervision, Writing - original draft, Writing - review and editing

## Author ORCIDs

Dawei Cai http://orcid.org/0000-0001-6650-0217
Siqi Zhu http://orcid.org/0000-0001-6307-1189
Mian Gong http://orcid.org/0000-0002-1785-0621
Yu Jiang http://orcid.org/0000-0003-4821-3585

## Decision letter and Author response

Decision letter https://doi.org/10.7554/eLife.73346.sa1
Author response https://doi.org/10.7554/eLife.73346.sa2

---

# Additional files

## Supplementary files

• Supplementary file 1. Tables that support the analysis and results above. (a) Sample information. Dates are estimated from either calibrated radiocarbon dating (bold) or from the archaeological context. Sex is inferred from the ratio of depth of coverage found on the X chromosome and autosomes (F, female; M, male) (c), and the average depth of coverage when mapping against both of the horse and donkey reference genomes after rescaling and trimming are provided. (b) Calibrated radiocarbon measurement summary statistics and dating of five ancient horses sequenced in this study. Uncal BP dates were calibrated using OxCalOnline (https://c14.arch.ox.ac.uk/oxcal.html) with the IntCal20 calibration curve. (c) Sex information. The mean coverage of the autosomes and the X chromosome together with the ratio between them (F, female; M, male). (d) Comparative Genome Panel. (e) Mitochondrial sequences used in this study. (f) Variance explained by TreeMix models from 0 to 3 migration edges excluding transitions. Monotonic increase of the variance explained by the model stopped when considering more than 3 migration edges. (g) Inference of total migration rates ($M$) and migration proportions ($p$) using G-PhoCS (*Gronau et al., 2011*). A total of five models, including various possible migration bands, were considered. Models 1–4 include migration bands between *E. ovodovi* and other lineages, while model 5 contains all gene flow events identified in *Jónsson et al., 2014*. The migration bands with significant gene flow are highlighted in bold (these were defined as having a mean value of $M > 3\%$ and 95% credible interval not intercepting 0). They were combined to establish the final demographic model shown in *Figure 3*. (h). Migration rate estimates returned by G-PhoCS. The 95% credible intervals of four significant migration bands identified in (g) are shown. (i) Parameter estimates returned by G-PhoCS, considering models with and without migrations. The topology is in the form of (*E. caballus*, (*E. ovodovi*, ((*E. a. somalicus*, *E. a. africanus*), (*E. kiang*, *E. hemionus*)))), (((*E. b. quagga*, *E. b. boehmi*), *E. grevyi*), *E. z. hartmannae*). Divergence time and population size are estimated by the 95% Bayesian credible interval using total 15,324 candidate 'neutral' loci, considering the sequence data aligned against the horse reference genome. The migration model contains the four significant migration bands estimated and provided in (h). (j) The tip dates (average calibrated radiocarbon dates or dates were estimated from the archaeological context) for sample ages in BEAST analyses.

• Transparent reporting form

## Data availability

Sequencing data have been deposited in the European Nucleotide Archive under the accession number PRJEB44527.

The following dataset was generated:

| Author(s) | Year | Dataset title | Dataset URL | Database and Identifier |
|---|---|---|---|---|
| Zhu S, Gong M, Zhang N, Wen J, Liang Q, Sun W, Shao X, Guo Y, Cai Y, Zheng Z, Zhang W, Hu S, Wang X, Tian H, Li Y, Liu W, Yang M, Yang J, Wu D, Orlando L, Jiang Y, Cai D | 2021 | Our sequence data provided 26 mitochondrial genomes and 3 complete nuclear genomes for Equus (Sussemionus) ovodovi | https://www.ebi.ac.uk/ena/browser/view/PRJEB44527?show=reads | European Nucleotide Archive, PRJEB44527 |

The following previously published datasets were used:

| Author(s) | Year | Dataset title | Dataset URL | Database and Identifier |
|---|---|---|---|---|
| Renaud G, Petersen B, Seguin-Orlando A, Bertelsen M F, Waller A, Newton R, Paillot R, Bryant N, Vaudin M, Librado P, Orlando L | 2018 | This study aims at improving the genome reference of the domestic donkey using the Chicago/HiRiSe technology | https://www.ebi.ac.uk/ena/browser/view/PRJEB24845?show=reads | European Nucleotide Archive, PRJEB24845 |
| Jonsson H | 2014 | Speciation with gene flow in equids despite extensive chromosomal plasticity | https://www.ebi.ac.uk/ena/browser/view/PRJEB7446?show=reads | European Nucleotide Archive, PRJEB7446 |
| Ginolhac A | 2013 | General Sample for Equus asinus asinus, Willy | https://www.ncbi.nlm.nih.gov/biosample/?term=SAMN02179859 | NCBI BioSample, SAMN02179859 |
| Dugarjaviin M | 2014 | Model organism or animal sample from Equus hemionus | https://www.ncbi.nlm.nih.gov/biosample/?term=SAMN03010637 | NCBI BioSample, SAMN03010637 |
| Kalbfleisch TS | 2014 | Sample from Equus caballus | https://www.ncbi.nlm.nih.gov/biosample/?term=SAMN02953672 | NCBI BioSample, SAMN02953672 |
| Achilli A | 2012 | Mitochondrial genomes from modern horses reveal the major haplogroups that underwent domestication | https://www.ncbi.nlm.nih.gov/nuccore/347361635/ | NCBI GenBank, 347361635 |
| Vilstrup JT | 2013 | Mitochondrial phylogenomics of modern and ancient equids | https://www.ncbi.nlm.nih.gov/nuccore/JX312719 | NCBI GenBank, JX312719 |
| Vilstrup JT | 2013 | Mitochondrial phylogenomics of modern and ancient equids | https://www.ncbi.nlm.nih.gov/nuccore/JX312721 | NCBI GenBank, JX312721 |
| Vilstrup JT | 2013 | Mitochondrial phylogenomics of modern and ancient equids | https://www.ncbi.nlm.nih.gov/nuccore/JX312725 | NCBI GenBank, JX312725 |
| Vilstrup JT | 2013 | Mitochondrial phylogenomics of modern and ancient equids | https://www.ncbi.nlm.nih.gov/nuccore/JX312730 | NCBI GenBank, JX312730 |
| Vilstrup JT | 2013 | Mitochondrial phylogenomics of modern and ancient equids | https://www.ncbi.nlm.nih.gov/nuccore/JX312732 | NCBI GenBank, JX312732 |
| Vilstrup JT | 2013 | Mitochondrial phylogenomics of modern and ancient equids | https://www.ncbi.nlm.nih.gov/nuccore/JX312734 | NCBI GenBank, JX312734 |

*Continued*

| Author(s) | Year | Dataset title | Dataset URL | Database and Identifier |
|---|---|---|---|---|
| Der Sarkissian C | 2015 | Mitochondrial genomes reveal the extinct Hippidion as an outgroup to all living equids | https://www.ncbi.nlm.nih.gov/nuccore/KM881671 | NCBI GenBank, KM881671 |
| Libradoa P | 2015 | Tracking the origins of Yakutian horses and the genetic basis for their fast adaptation to subarctic environments | https://www.ncbi.nlm.nih.gov/nuccore/KT368725 | NCBI GenBank, KT368725 |
| Orlando L | 2016 | Recalibrating Equus evolution using the genome sequence of an early Middle Pleistocene horse | https://www.ncbi.nlm.nih.gov/nuccore/KT757740 | NCBI GenBank, KT757740 |
| Orlando L | 2016 | Recalibrating Equus evolution using the genome sequence of an early Middle Pleistocene horse | https://www.ncbi.nlm.nih.gov/nuccore/KT757741 | NCBI GenBank, KT757741 |
| Druzhkova AS, Makunin AI, Vorobieva NV, Vasiliev SK, Ovodov ND, Shunkov MV, Trifonov VA, Graphodatsky AS | 2017 | Complete mitochondrial genome of an extinct Equus (Sussemionus) ovodovi specimen from Denisova cave (Altai, Russia) | https://www.ncbi.nlm.nih.gov/nuccore/KY114520 | NCBI GenBank, KY114520 |
| Heintzman PD, Zazula GD, MacPhee R, Scott E, Cahill JA, McHorse BK, Kapp JD, Stiller M, Wooller MJ, Orlando L, Southon J, Froese DG, Shapiro B | 2018 | A new genus of horse from Pleistocene North America | https://www.ncbi.nlm.nih.gov/nuccore/MF134655 | NCBI GenBank, MF134655 |
| Xu X, Gullberg A, Arnason U | 2016 | The complete mitochondrial DNA (mtDNA) of the donkey and mtDNA comparisons among four closely related mammalian species-pairs | https://www.ncbi.nlm.nih.gov/nuccore/X97337 | NCBI GenBank, X97337 |

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
