## [Editor Report]

This article represents multiple milestones in our understanding of the evolution and extinction of Pleistocene equids, including revising the timing of extinction and clarifying the evolutionary history of *Equus* (*Sussemionus*) *ovodovi*. The discovery of the late persistence of non-caballine equid taxa in Northern China until deep into the late Holocene is particularly important. This finding will be of broad interest to the paleontology, paleoecology, archaeology, and paleogenomic communities and should stimulate important future research into equid extinction processes.

---

## [Decision Letter]

**Decision letter after peer review:**

Thank you for submitting your article "Ancient genomes redate the extinction of *Sussemionus*, a subgenus of *Equus*, to late Holocene" for consideration by *eLife*. Your article has been reviewed by 2 peer reviewers, and the evaluation has been overseen by a Reviewing Editor and George Perry as the Senior Editor. The reviewers have opted to remain anonymous.

Essential Revisions include providing additional detail on existing datasets, the adjustment and re-processing of some analyses, and the addition of a section on archaeological context and expanded discussion on regional zooarchaeological implications of your findings. These points and more excellent feedback are detailed in the below reviews. In addition, I agree with reviewer #2s suggestion to revise your title for the next version of your paper.

Overall: Well done on this paper. We look forward to receiving and reviewing your revision. I will share it with the two reviewers to confirm that their comments have been suitably addressed.

*Reviewer #2 (Recommendations for the authors):*

I suggest the author's reconsider their manuscript title as it is the radiocarbon dates, rather than the ancient genomes, that re-date the extinction.

Given the specimens under study are isolated skeletal elements from a handful of stratigraphic horizons, it is possible that some of the specimens belonged to the same individual. There should be a description in the Methods about the minimum number of individuals at each locality. The mitogenomic data would prove valuable in this regard.

The authors use relative X-chromosome and autosomal coverage to distinguish male and female individuals (L104-106). However, these assignments should be justified with data. A new supplementary table with mean coverage of the autosomes and the X chromosome together with the ratio between the two would suffice.

There are issues with the BEAST-derived Bayesian phylogeny (Figure 2 —figure supplement 6). First, the ages of the tips do not appear to have been constrained with the known ages of the ancient specimens (or they are out by an order of magnitude). For example, JX312734 looks to be ~400,000 years old, when its age (based on Table S4) is 40,000 years old. Note also that ancient E. caballus individuals have been constrained as modern. Second, the constant population and strict clock models used (L538) are not suitable for the interspecies analysis used here. The authors should consider the Birth-death serially-sampled and relaxed clock models. Third, there is no description of how the molecular clock was calibrated (fossil, previous genomic estimate, and/or fixed mutation rate). I refer the authors to two publications for reference (dois: 10.1111/mec.15977, 10.7554/*eLife*.29944).

The authors present two D-statistics analyses based on alignment to either the horse or donkey reference genomes, which give very different results (Figure 3 —figure supplement 1 and 2). Presumably the larger D-stats when an African ass is included in the donkey reference analysis is an artifact of reference genome bias, and so the horse genome (outgroup) results should be considered more reliable. The authors should add a statement about this disagreement and an explanation in the Results, as this will not be clear to non-specialists, especially given the statement on L579-581 that the donkey reference genome analysis was included to 'eliminate the bias of the reference genome'.

It is stated that Z-scores of {greater than or equal to}3 were considered statistically significant on L577-579. However, the Z-score data are not presented and so it is not possible to determine which of the D-statistics in Figure 3 —figure supplement 1 and 2 are significant or not (L793, L801).

The G-PhoCS analysis suggests major introgression between the early non-caballine equid lineages. However, none of these events are recovered in the TreeMix analyses even with up to three migration edges considered. The apparently conflicting signal between these two analyses needs to be explained.

It will not be clear to non-specialists how the total migration rate for a single direction can be >100% (Tables S6 and S7). Please add a statement in the Methods as to why this occurs.

More details are needed in the G-PhoCs methods to enable reproducibility. Specifically, how were non-coding RNA genes identified and removed (L599-600), and what thresholds and methods were used to detect enriched misaligned bases and high false negative rates (L602-604)? In this regard, it may be helpful if the authors made their code available for these methods.

There is no discussion of the discordance between the mitochondrial and nuclear genome trees, which the G-PHoCS analysis seems to shed some light upon. I invite the authors to comment on this.

I almost missed that the authors have already made their raw sequence data publicly available (included in *eLife* manuscript information but not in the manuscript). To ensure readers can easily find the raw data, I suggest that the authors give a link to the European Nucleotide Archive BioProject code on L456.

[Editors' note: further revisions were suggested prior to acceptance, as described below.]

Thank you for resubmitting your work entitled "Ancient DNA research redates the extinction of *Sussemionus*, a subgenus of *Equus*, to late Holocene" for further consideration by *eLife*. Your revised article has been evaluated by George Perry (Senior Editor) and the two reviewers of the previous version of your paper.

The manuscript has been improved but there are some remaining issues that need to be addressed. Detailed points of required revision are noted below, but in addition I will note that in my view your manuscript revisions in response to excellent and important points made by the reviewers in their original review are too superficial in multiple instances, e.g. in cases where expanded discussion or incorporation of particular concepts into your interpretation were requested, but the points were addressed with the addition of a short sentence only rather than taking the opportunity to maximally improve the manuscript, which is what we expect.

Thus, in your response, please detail the further revisions you made to the previous set of review comments, with the above in mind, in addition to further point-by-point responses to the specific comments below. This will be the final opportunity to revise your manuscript.

1. The issue with the title is not resolved.

2. Please provide expanded information on the Β Analytic 14C methods and results, if at all possible. (please contact the company for more specific information on how the samples were processed, for the sake of methodological completeness and data reproducibility).

3. The level of detail in the new archaeological background paragraph should be further improved. The revisions do accomplish the important goal of pointing the reader to the relevant background literature/citations, but acknowledgment and/or summary of the state of knowledge of the archaeological record of equids in the study region is incomplete. At the least this should include reference to Yuan and Flad's 2006 summary (Research on Early Horse Domestication in China. In Equids in Time and Space, ed. by Marjan Mashkour, pp. 124-131.)

4. The new sentence on 102-103 should be removed, and the sentence ending this paragraph on lines 104-105 needs to explain much more specifically what "no traces of domestication" means (e.g. no paleopathological problems?) and what "indicates they were hunted for food" (e.g. butchery patterns indicating meat removal? arrowheads imbedded in bone?).

5. It is unclear where the information about minimum number of individuals is coming from. The authors state that there were at least "31 individuals in the Honghe samples" (L460), yet there are only 20 samples from this locality in Supplementary File 1a. Further, the authors need to expand on the statement "the same process is repeated for the other two sites to ensure the specimens are unique" by including comparable counts.

6. We thank the authors for applying some of the suggested changes to the BEAST analysis. However, the tip dates for known sample ages have still not been constrained. Contrary to the rebuttal letter, there should not be any "deviations from the known ages" as these parameters should be fixed.

7. The explanation for the discordance between the G-PhoCS and TreeMix analyses needs to be stated in the manuscript.

8. G-PhoCS burn-in: although the MCMC run settings may use a burn-in of 0, the burn-in needs to be removed during post-processing (as is stated in the G-PhoCS user manual). For example, Vershinina et al. 2021 (doi: 10.1111/mec.15977) used a burn-in of 10%. This needs to be applied.

9. Figure 4 figure supplement 2: This is still confusing to the reader. The panel keys should be updated to reflect that all PSMC plots are based on E. a. somalicus.

10. Supplementary File 1c: correct 'gender' to 'sex'

11. Supplementary File 1e: correct 'yBC'. The age given for Haringtonhippus francisci is uncalibrated.

---

## [Author Response]

Reviewer #2 (Recommendations for the authors):I suggest the author's reconsider their manuscript title as it is the radiocarbon dates, rather than the ancient genomes, that re-date the extinction.

Thank you for your suggestion, and we have revised the title accordingly: “Ancient DNA research redates the extinction of *Sussemionus*, a subgenus of *Equus*, to late Holocene”.

Given the specimens under study are isolated skeletal elements from a handful of stratigraphic horizons, it is possible that some of the specimens belonged to the same individual. There should be a description in the Methods about the minimum number of individuals at each locality. The mitogenomic data would prove valuable in this regard.

We thank the reviewer for pointing this out. The description about the minimum number of individuals was added in the Methods (lines 457-461): “Considering the preservation status and quantity, the minimum number of individuals was determined by assigning the frequency of hip bone and was calculated from the acetabular bone to avoid double-counting. Based on counts of skeletal elements, there is a minimum of 31 individuals in the Honghe samples. The same process is repeated for the other two sites to ensure the specimens are unique”. Meanwhile, the mitochondrial genome of each individual was visually checked to ensure they are unique according to your suggestion.

The authors use relative X-chromosome and autosomal coverage to distinguish male and female individuals (L104-106). However, these assignments should be justified with data. A new supplementary table with mean coverage of the autosomes and the X chromosome together with the ratio between the two would suffice.

We apologize for the lack of information here. We have now added Supplementary File 1c with mean coverage of the autosomes and the X chromosome together with the ratio between them.

There are issues with the BEAST-derived Bayesian phylogeny (Figure 2 —figure supplement 6). First, the ages of the tips do not appear to have been constrained with the known ages of the ancient specimens (or they are out by an order of magnitude). For example, JX312734 looks to be ~400,000 years old, when its age (based on Table S4) is 40,000 years old. Note also that ancient E. caballus individuals have been constrained as modern. Second, the constant population and strict clock models used (L538) are not suitable for the interspecies analysis used here. The authors should consider the Birth-death serially-sampled and relaxed clock models. Third, there is no description of how the molecular clock was calibrated (fossil, previous genomic estimate, and/or fixed mutation rate). I refer the authors to two publications for reference (dois: 10.1111/mec.15977, 10.7554/eLife.29944).

Thank you for this suggestion. First, we have set tip dates according to the ages of species in Supplementary File 1e. But considering an enormous range of time scales, few ancient specimens may showed some deviations from the known ages in Bayesian phylogeny.

Second, we have reconstructed Bayesian phylogeny using Birth-death model and relaxed molecular clock in Figure 2—figure supplement 4 according to your suggestion.

Third, we have added the description in lines 602-604: “we calibrated the tree using an age of 4–4.5 Mya for the root of crown group E. caballus (normal prior, mean 4.25 Mya, stdev: 0.15 Mya)” (see L. Orlando et al. (2013), https://www.nature.com/articles/nature12323).

The authors present two D-statistics analyses based on alignment to either the horse or donkey reference genomes, which give very different results (Figure 3 —figure supplement 1 and 2). Presumably the larger D-stats when an African ass is included in the donkey reference analysis is an artifact of reference genome bias, and so the horse genome (outgroup) results should be considered more reliable. The authors should add a statement about this disagreement and an explanation in the Results, as this will not be clear to non-specialists, especially given the statement on L579-581 that the donkey reference genome analysis was included to 'eliminate the bias of the reference genome'.

Thanks for spotting this. The similar statement was given on lines 647-649 in the manuscript, and we have added the sentence “Given the larger D-stats when an African ass is included in the donkey reference analysis is an artifact of reference genome bias, so that the horse reference genome results should be considered more reliable” in lines 649-652 according to the suggestion.

It is stated that Z-scores of {greater than or equal to}3 were considered statistically significant on L577-579. However, the Z-score data are not presented and so it is not possible to determine which of the D-statistics in Figure 3 —figure supplement 1 and 2 are significant or not (L793, L801).

We apologize for missing the legends to present the Z-score data. We have added the sentences “The nonsignificant results are shown in gray” in lines 887-888 and 895-896.

The G-PhoCS analysis suggests major introgression between the early non-caballine equid lineages. However, none of these events are recovered in the TreeMix analyses even with up to three migration edges considered. The apparently conflicting signal between these two analyses needs to be explained.

We thank the reviewer for the suggestion. Previous studies found that the TreeMix models will work best when gene flow between populations is restricted to a relatively short time period, situations of continuous migration violate this assumption and lead to unclear results (see Pickrell, Joseph K., and Pritchard, Jonathan K. (2012), https://journals.plos.org/plosgenetics/article?id=10.1371/journal.pgen.1002967). So compared with the G-PhoCS analysis, the TreeMix analyses had its limitation in an enormous range of time scales.

It will not be clear to non-specialists how the total migration rate for a single direction can be >100% (Tables S6 and S7). Please add a statement in the Methods as to why this occurs.

Thank you for this suggestion. We have now added a statement in the Methods. “The total migration rate gives an accumulated rate over a long period of time so that it can be >100%.” (lines 714-716)

More details are needed in the G-PhoCs methods to enable reproducibility. Specifically, how were non-coding RNA genes identified and removed (L599-600), and what thresholds and methods were used to detect enriched misaligned bases and high false negative rates (L602-604)? In this regard, it may be helpful if the authors made their code available for these methods.

Thanks for the suggestion. The non-coding RNA genes were identified using GFF annotation files.

We apologize for being unclear, and we have changed the sentence in lines 673-677: “Besides the various hard filters described above, regions/sites likely to be enriched for misaligned bases, and to have high false negative rates during read alignment or variant detection were masked as missing data. So in this case, different individuals may be treated differently depending on the result of genotyping in section “Variant calling” depending on the presence of (1) indels, (2) triallelic sites, (3) positions with depth of coverage twice the mean depth recorded for each individual, and; (4) transition sites”. Enriched misaligned bases and high false negative rates were embodied in (1) indels, (2) triallelic sites, (3) positions with depth of coverage twice the mean depth recorded for each individual, and; (4) transition sites.

There is no discussion of the discordance between the mitochondrial and nuclear genome trees, which the G-PHoCS analysis seems to shed some light upon. I invite the authors to comment on this.

This is certainly interesting suggestion. The discordance between the mitochondrial and nuclear genome trees can be caused by two reasons. First, mitochondrial DNA is maternally inherited and therefore variation in it will reflect disper-sal and history of the maternal lineage only. Second, two mitochondrial Maximum Likelihood trees based on all 6 partitions and excluding the control region were both reconstructed in Figure 2—figure supplement 5. It is not the latter but the former is discordant with nuclear genome trees, which may cause by exhibited significant increased damage in the mitochondrial control region.

I almost missed that the authors have already made their raw sequence data publicly available (included in eLife manuscript information but not in the manuscript). To ensure readers can easily find the raw data, I suggest that the authors give a link to the European Nucleotide Archive BioProject code on L456.

Thanks for the suggestion, and we have now given a link to the European Nucleotide Archive BioProject code in line 512.

[Editors' note: further revisions were suggested prior to acceptance, as described below.]

The manuscript has been improved but there are some remaining issues that need to be addressed. Detailed points of required revision are noted below, but in addition I will note that in my view your manuscript revisions in response to excellent and important points made by the reviewers in their original review are too superficial in multiple instances, e.g. in cases where expanded discussion or incorporation of particular concepts into your interpretation were requested, but the points were addressed with the addition of a short sentence only rather than taking the opportunity to maximally improve the manuscript, which is what we expect.Thus, in your response, please detail the further revisions you made to the previous set of review comments, with the above in mind, in addition to further point-by-point responses to the specific comments below. This will be the final opportunity to revise your manuscript.1. The issue with the title is not resolved.

We have rephrased the title to indicate that it is the combination of both radiocarbon dating and phylogenomic that help reconsider the extinction/survival of *Equus Sussemionus* to the late Holocene. Our new title reads as follows:

“Radiocarbon and genomic evidence for the survival of *Equus Sussemionus* until the late Holocene”

2. Please provide expanded information on the Β Analytic 14C methods and results, if at all possible. (please contact the company for more specific information on how the samples were processed, for the sake of methodological completeness and data reproducibility).

We now provide the requested information as:

– A dedicated paragraph in the Methods section (page 29, lines 555-560: “Radiocarbon dating of the samples was performed at the Β Analytic Radiocarbon Dating Laboratory, Miami, Florida. Bone or tooth pieces about 2g were sampled in the bone and sent for subsequent dating of collagen (not ultrafiltered). Calibration was carried out using OxCalOnline (https://c14.arch.ox.ac.uk/oxcal.html) and the IntCal20 calibration curve. Calibrated dates are provided in Supplementary File 1b.”);

– A sentence in the main text (lines 142-146): “Combined, these samples were radiocarbon dated to 3,456-4,460 calibrated years before the present (cal BP), including a mid-second millennium BCE date for the most recent sample, HH13H (3270±30 uncal. BP, i.e. 3,456-3,616 cal BP) (Supplementary File 1b).”

– A table referring to laboratory numbers, uncalibrated estimates and confidence range, and calendar years calibrated estimates (IntCal20) (see Supplementary File 1b).

3. The level of detail in the new archaeological background paragraph should be further improved. The revisions do accomplish the important goal of pointing the reader to the relevant background literature/citations, but acknowledgment and/or summary of the state of knowledge of the archaeological record of equids in the study region is incomplete. At the least this should include reference to Yuan and Flad's 2006 summary (Research on Early Horse Domestication in China. In Equids in Time and Space, ed. by Marjan Mashkour, pp. 124-131.)

We apologize for being unclear. Based on all excavated equine fossil bones found at Honghe, the Minimum Number of Individuals (NMI) was estimated to 31 individuals. And because of the preservation status, ancient DNA sequences were recovered from 20 of the 31 samples (Supplementary File 1a). This is now fully detailed at page 26 (lines 491-501): “Minimum number of individuals (MNI) was determined by assigning the frequency of hip bone and was calculated from the acetabular bone to avoid double-counting. MNI was estimated to 31 individuals at Honghe, 4 at Muzhuzhuliang and 4 at Shatangbeiyuan. DNA preservation conditions were compatible with the recovery of ancient DNA sequences from only 20 of the 31 Honghe samples, 3 of the 4 Muzhuzhuliang samples, and 3 of the 4 Shatangbeiyuan samples (Supplementary File 1a).”

4. The new sentence on 102-103 should be removed, and the sentence ending this paragraph on lines 104-105 needs to explain much more specifically what "no traces of domestication" means (e.g. no paleopathological problems?) and what "indicates they were hunted for food" (e.g. butchery patterns indicating meat removal? arrowheads imbedded in bone?).

We have now removed the sentence indicated, and have rephrase the following ones, appearing on lines 117-125: “No obvious signs of domestication, including paleopathologies related to horseback riding, bridling or chariotry (Bendrey, 2007; Taylor and Tuvshinjargal 2018), were found amongst the equine specimens investigated at the three sites. In contrast, slash marks could be identified on some of the bones (HH13H, HH26H, and MZ104H), together with indications of bone marrow extraction (Figure 1—figure supplement 2). These findings suggest these specimens were hunted.”

5. It is unclear where the information about minimum number of individuals is coming from. The authors state that there were at least "31 individuals in the Honghe samples" (L460), yet there are only 20 samples from this locality in Supplementary File 1a. Further, the authors need to expand on the statement "the same process is repeated for the other two sites to ensure the specimens are unique" by including comparable counts.

We apologize for being unclear. Based on all excavated equine fossil bones found at Honghe, the Minimum Number of Individuals (NMI) was estimated to 31 individuals. And because of the preservation status, ancient DNA sequences were recovered from 20 of the 31 samples (Supplementary File 1a). This is now fully detailed at page 26 (lines 491-501): “Minimum number of individuals (MNI) was determined by assigning the frequency of hip bone and was calculated from the acetabular bone to avoid double-counting. MNI was estimated to 31 individuals at Honghe, 4 at Muzhuzhuliang and 4 at Shatangbeiyuan. DNA preservation conditions were compatible with the recovery of ancient DNA sequences from only 20 of the 31 Honghe samples, 3 of the 4 Muzhuzhuliang samples, and 3 of the 4 Shatangbeiyuan samples (Supplementary File 1a).”

6. We thank the authors for applying some of the suggested changes to the BEAST analysis. However, the tip dates for known sample ages have still not been constrained. Contrary to the rebuttal letter, there should not be any "deviations from the known ages" as these parameters should be fixed.

We have proceeded according to the editor’s suggestion and have now used tip-calibrations (average calibrated radiocarbon dates) in our BEAST analyses (Supplementary File 1j). The full procedure is now described on lines 633-649, with the resulting tree shown on Figure 2—figure supplement 4.

7. The explanation for the discordance between the G-PhoCS and TreeMix analyses needs to be stated in the manuscript.

We have added the requested explanation in lines 787-805 (pages 39-40): “We caution that the analyses carried out using TreeMix and G-PhoCS returned partly discordant results. This may be due to TreeMix modelling pulses of admixture in contrast to G-PhoCS, in which situations of continuous gene flow can be accommodated. Additionally, gene flow affecting the two deepest tree branches can be directly accommodated by reducing their divergence. Therefore, the deep admixture inferred by G-PhoCS from the caballine branch into the ancestral branch of Sussemiones and other non-caballine equids cannot be expected to be identified through an individual migration edge with TreeMix, as this could simply be modelled through a more limited divergence between both underlying lineages. The same holds true for the asymmetric gene flow inferred by G-PhoCS between Sussemiones and the branch ancestral to all extant asses; TreeMix is likely to only identify the resulting unidirectional contribution of these admixtures, which mainly sources to the branch ancestral to extant asses into Sussemiones; since G-PhoCS also infers additional admixture from the branch ancestral to extant zebras into Sussemiones, we can expect TreeMix to accommodate both sources of gene flow through a reduced divergence between Sussemiones and the branch ancestral to stenonines. Finally, it may reflect limitations pertaining to the two underlying data sets utilized, consisting on the one hand to the whole-genome SNP panel for TreeMix, further filtered for 15,324 candidate ‘neutral’ loci in G-PhoCS.”

8. G-PhoCS burn-in: although the MCMC run settings may use a burn-in of 0, the burn-in needs to be removed during post-processing (as is stated in the G-PhoCS user manual). For example, Vershinina et al. 2021 (doi: 10.1111/mec.15977) used a burn-in of 10%. This needs to be applied.

We apologize for the unclear explanation. Although we run the MCMC with a pre-set burn-in of 0, the first 10% iterations were removed as burn-in during post-processing with Tracer v1.6 (http://tree.bio.ed.ac.uk/software/tracer/). This is shown in Author response image 1. Accordingly, we have edited the sentence in lines 742-743 (page 37): “The final results in Figure 3 are based on 500,000 MCMC iterations, considering the first 10% as burn-in.”

**Author response image 1. sa2fig1:** 

9. Figure 4 figure supplement 2: This is still confusing to the reader. The panel keys should be updated to reflect that all PSMC plots are based on E. a. somalicus.

We apologize for being unclear. The figure captions have now been rephrased to describe the procedure followed. It reads as follows (pages 60-61, lines 961-969): “Figure 4—figure supplement 2. Determining the uniform false-negative rate (uFNR) that was necessary for scaling PSMC results. (A) HH06D (11.30×), (B) KIA (10.68×) and (C) ONA (18.38×). The most suitable uFNR values for rescaling the PSMC profile are reported between squared brackets, to the right of the species names considered. The PSMC trajectory retrieved when considering all the sequence data available for the SOM individual is shown in blue. The green line provides the PSMC trajectory reconstructed when down-sampling these data to the average genome depth of coverage obtained for the species examined (top: *Equus Sussemionus*, red; centre: *Equus kiang*, purple, and; bottom: *E. hemionus*, purple).”

10. Supplementary File 1c: correct 'gender' to 'sex'

Done.

11. Supplementary File 1e: correct 'yBC'. The age given for Haringtonhippus francisci is uncalibrated.

Done.